# Humidity-dependent lubrication of highly loaded contacts by graphite and a structural transition to turbostratic carbon

Carina Elisabeth Morstein [1,2,6], Andreas Klemenz [2,6], Martin Dienwiebel [1,2] ✉ & Michael Moseler [2,3,4,5] ✉

Graphite represents a promising material for solid lubrication of highly loaded tribological contacts under extreme environmental conditions. At low loads, graphite's lubricity depends on humidity. The adsorption model explains this by molecular water films on graphite leading to defect passivation and easy sliding of counter bodies. To explore the humidity dependence and validate the adsorption model for high loads, a commercial graphite solid lubricant is studied using microtribometry. Even at 1 GPa contact pressure, a high and low friction regime is observed - depending on humidity. Transmission electron microscopy reveals transformation of the polycrystalline graphite lubricant into turbostratic carbon after high and even after low load (50 MPa) sliding. Quantum molecular dynamics simulations relate high friction and wear to cold welding and shear-induced formation of turbostratic carbon, while low friction originates in molecular water films on surfaces. In this work, a generalized adsorption model including turbostratic carbon formation is suggested.

Graphite is one of the oldest technically used solid lubricants[1]. Since the discovery of its lamellar structure via X-ray diffraction by Bragg in 1928, the lubricating effect of graphite has often been described in terms of mutual sliding of its basal planes[2] (see left scheme in Fig. 1a). This illustrative sliding mechanism is often referred to as the deck-of-cards[3] or lattice-shear model[4] and is still very popular today to explain the lubricating effect of graphite[5–7].

First doubts about this simple explanation arose as early as the 1930s, when reports emerged that friction and wear of graphite contacts increase drastically at low humidity[8]. Savage showed that graphite exhibits good lubricity only in humid atmospheres and in some gases, and is not intrinsically a good lubricant[9]. Since the original lattice-shear model fails to explain this effect, Rowe assumed that intercalated water or gas molecules could increase the layer distance, thus reducing friction[10]. An accordingly modified lattice-shear model is

depicted in the right scheme of Fig. 1a. However, intercalation could be excluded by X-ray diffraction experiments looking for water in bulk[11] and between terminating graphene layers on top of graphite[4]. Micro-tribometer experiments with a W stylus on the basal plane of graphite provided further evidence for the failure of the lattice-shear model[12].

Bollmann and Spreadborough proposed a roller mechanism[13]: When moved laterally, individual packets of graphene layers roll together to form graphene scrolls that lubricate via a bearing-like mechanism (Fig. 1c). Since the influence of humidity is explained by water intercalation (similar to Rowe[10]), the roller model is in disagreement with later experimental X-ray characterization as well[11].

The adsorption model[9] represents an alternative explanation for the lubricity of graphite (Fig. 1b). Its basic form dates back to the seminal work of Savage[9], who suggested that molecular water layers adsorbed on top of the graphite act as boundary lubrication films. In

[1]Karlsruhe Institute of Technology (KIT), IAM - Institute for Applied Materials, Straße am Forum 7, 76131 Karlsruhe, Germany. [2]Fraunhofer-Institute for Mechanics of Materials IWM, MicroTribology Center µTC, Wöhlerstraße 11, 79108 Freiburg, Germany. [3]Institute of Physics, University of Freiburg, Hermann-Herder-Straße 3, 79104 Freiburg, Germany. [4]Freiburg Materials Research Center, University of Freiburg, Stefan-Meier-Str. 21, 79104 Freiburg, Germany. [5]Cluster of Excellence livMatS, Freiburg Center for Interactive Materials and Bioinspired Technologies, University of Freiburg, Georges-Köhler-Allee 105, 79110 Freiburg, Germany. [6]These authors contributed equally: Carina Elisabeth Morstein, Andreas Klemenz. ✉e-mail: martin.dienwiebel@kit.edu; michael.moseler@iwm.fraunhofer.de

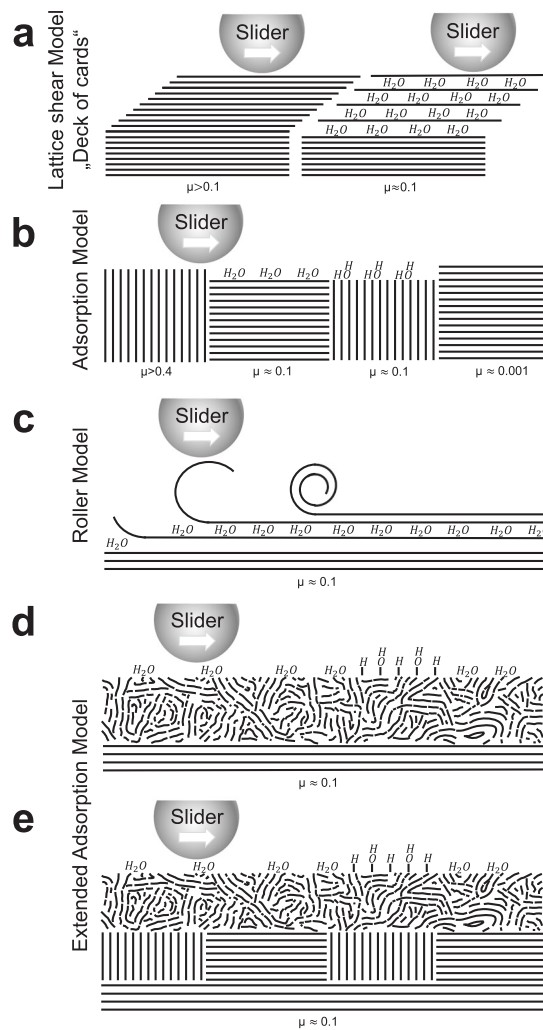

**Fig. 1 | Schematic representation of the different models for solid lubrication by graphite. a** Lattice-shear model: dry (left) and with water intercalation (right). **b** Adsorption model. **c** Roller model. **d**, **e** Adsorption model for graphite extended by shear-induced formation of turbostratic carbon for **d** high loads (≈1000 MPa) and **e** low loads (<100 MPa). Typical coefficients of friction $\mu$ values are given under the respective schematic drawing.

results in increased friction, while adsorbed water passivates defects and conserves graphite's lubricity.

Graphite can also serve as a solid lubricant in high-temperature grinding[20] or in wheel/rail systems[21]. In such highly loaded tribological contacts, the applicability of the adsorption model is questionable, since the stability of water layers and graphite crystals might be compromised under high normal pressures. Clearly, a better mechanistic understanding of graphite lubrication under high loads would help to design long-lived tribological parts that operate in extreme environments.

In this work, the influence of humidity on graphite lubrication under high loads is investigated experimentally at the macroscale using microtribometer experiments and at the atomic scale using density functional based tight-binding (DFTB) molecular dynamics (MD) simulations. In both, a structural transition of the graphite crystals to t-C is observed. At sufficiently high humidity, the formation of the t-C tribolayer is accompanied by an increase of lubricity. The atomistic simulations attribute this effect to molecular water films—in agreement with the adsorption model. However, our results suggest that this model has to be extended by shear-induced t-C formation and the low friction sliding on water films supported by t-C (Fig. 1d). Additional experiments showing t-C formation at the sliding interface already at 50 MPa contact pressure indicate that our extended adsorption model applies also for low loads (Fig. 1e).

## Results
### Experiments
Graphite solid lubrication layers with a 3.5 μm average thickness are prepared by coating polished iron plates with a commercial graphite suspension using an airbrush gun, followed by a two day drying period to remove the solvent (details in the "Methods" section). The coating prepared in this way is an excellent model material for a technical graphite solid lubricant. To analyze the influence of the humidity on graphite lubrication, friction between the graphite-coated iron plate and a 100Cr6 steel sphere is measured with a linear-reversing microtribometer (see scheme in Fig. 2a) with a Hertz pressure $P_{Hertz}$ of 1 GPa or 50 MPa. The relative humidity (RH) is varied between values below 5% RH (dry pressurized air) and 45% RH using a self-built humidity controller; plus an extreme experiment, where a drop of deionized water was put into the tribological contact, leading to complete immersion. Frictional forces are measured during 500 cycles and the coefficient of friction (CoF, $\mu$) is calculated by dividing through the current applied normal force.

Figure 2b displays the evolution of the CoF for various relative humidities. After a short running-in period (lasting less than 80 cycles) the CoF remains low and stable for the experiments conducted under RH ≤ 30%. For higher humidity values (37% and 45% RH), an intermediate low friction period that ends with a steep rise of the CoF is observed after running in. The length of the stable low friction phase is referred to as $\tau_{low}$ and is defined as the number of cycles before the CoF exceeds the average of the previous ten cycles by 30%. For high humidities, a pronounced scatter of $\tau_{low}$ for different repetitions of the same experiment is observed (e.g., $\tau_{low}$ = 182, 280, and 440 cycles for RH = 37% and $\tau_{low}$ = 35, 205, and 220 cycles for RH = 45%). Despite this scatter, a reduction of the average $\tau_{low}$ with increasing RH is obvious (>500 for RH ≤ 30%, 300 for 37% RH, and $\tau_{low}$ = 153 for 45% RH). For the immersed contact no stable phase was observed.

Figure 2c displays the humidity dependence of the steady-state friction coefficients. Interestingly, the average low friction CoFs are slightly different for the individual RHs, indicating that an influence of surrounding humidity on friction is also present for high loads. An increase in humidity from ≤5% to 24% results in a slight decrease in friction from $\mu$ = 0.14 to 0.10, followed by a plateau with $\mu$ ≈ 0.1 for RH > 24%. Notably, an experiment with a reduced Hertz pressure of

addition, the adsorbed water functions as a hydrogen and oxygen reservoir for edge passivation of reactive graphene layers on basal planes[14,15] or in prismatic planes of graphite crystals perpendicular to the sliding direction[9,12].

Within the adsorption model, non-commensurate dry contacts between the basal planes of two graphite crystals[16] could lead to a further lowering of friction[15–17]. This implicitly assumes passivated graphite layers on both sliding surfaces. However, usually the solid lubricant initially only covers one tribopartner. In this case, transfer of the lubricant onto the counterbody is required. Transfer film formation has been previously observed, e.g., by Merkle & Marks[18] in transmission electron microscopy (TEM) experiments between graphitic flakes and a tungsten sliding probe or by Li et al. between a steel sphere and HOPG in a macroscopic contact in the first initial sliding cycles[19].

The adsorption model has been developed to explain graphite solid lubrication in low load technical applications, such as graphite-lubricated journal bearings or electrical contacts[9]. In these applications a macroscopic graphite layer consisting of a multitude of different graphite crystals is to be expected. It is assumed that during operation the collision between crystals lead to defect production and a steady change of the crystals' alignments. Under dry conditions, this

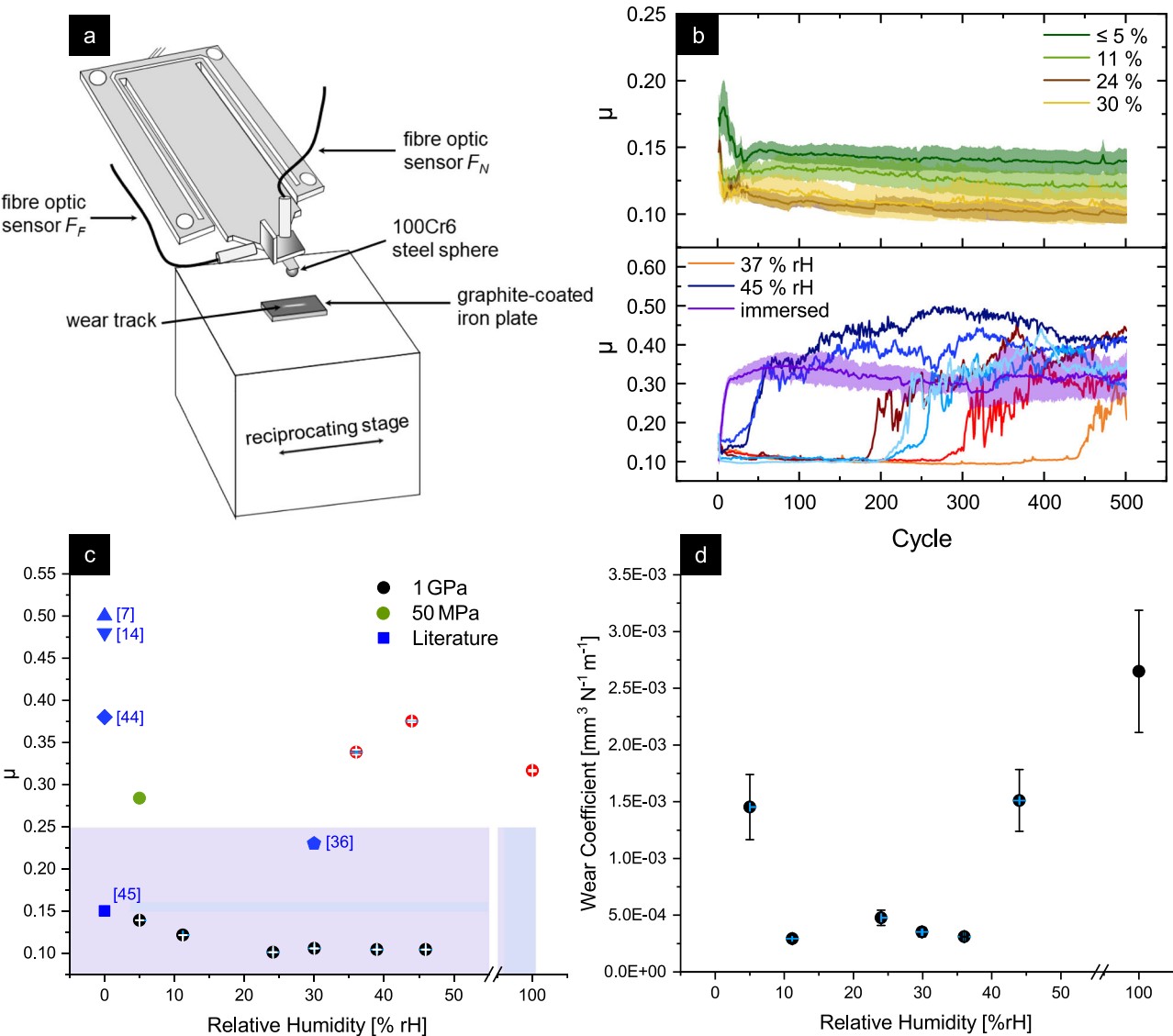

**Fig. 2 | Tribometric setup and experimental friction and wear results.**
**a** Schematic of the experimental setup. **b** Coefficient of friction ($\mu$) over the experimental duration for various humidity values. Each experiment was repeated three times for 500 cycles each at a stroke length of 1 mm, a velocity of 0.5 mm s$^{-1}$, and a normal force of 402 mN. For the experiments conducted at lower humidity RH ($\leq$30%) the average of three consecutive measurements for each humidity is shown in the upper panel. The standard deviation of three measurements is marked as the semi-transparent areas. In the lower panel all measurements at the higher humidity values (37% and 45% RH) are shown (shades of orange/red: 37% RH, shades of blue: 45% RH, purple: contact immersed in deionized water). **c** Average steady-state CoF after running-in with the standard error of the mean as error bars, and literature values[7,14,36,44,45] in blue as a comparison. Black circles represent the average of 100 cycles at the end of the experiment for RH $\leq$ 30% or the average of 20 cycles prior to rise in $\mu$ for RH > 30%. In the latter case, the steady-state high friction values are plotted in red (average of the last 100 cycles). **d** Wear coefficient of the graphite-coated iron plate after 500 cycles with standard error of the mean as error bars. Wear data of the counterbodies can be found in Supplementary Fig. 1.

50 MPa at RH $\leq$ 5% results in a CoF of 0.28 (green circle in Fig. 2b). This value is in line with previous experiments[22], where an increase in friction was observed with a decrease in normal force (e.g., at $P_{Hertz}$ = 250 MPa a $\mu$ = 0.29 was measured).

In addition, experimental literature values are shown as blue symbols in Fig. 2c with the range of typical CoF represented as a light blue area for experiments with no reported humidity. The friction measured in our study lies on the lower end of the typical CoFs, always yielding similar or even lower $\mu$ than comparable studies with graphite or graphene.

Wear coefficients of the plate as a function of RH are depicted in Fig. 2d. They are obtained by measuring the wear volume of the graphite-coated plates using confocal microscope topography (see Methods). Evidently, the wear coefficient is highest under dry pressurized air (RH $\leq$ 5%) and under high relative humidity values above

45%. At intermediate RH, the wear coefficient is an order of magnitude smaller showing only little variation.

Wear tracks are further analyzed using scanning electron microscopy; see Fig. 3. At low loads, wear on the graphite is barely visible (in Fig. 3a only flattening of the roughness peaks is seen). At high loads wear is much stronger and the size of the wear track exhibits a pronounced RH dependence. For RH $\leq$ 30%, (Fig. 3b, c), a thin carbon layer remains in the middle of the wear track, as seen in previous studies on substrates with industrial surface roughness values[22]. For high humidity (45% RH, Fig. 3d) this thin layer is worn away and adhesive wear is observed due to cold welding between the steel sphere and the iron plate. For the high and low humidity values (Fig. 3b, d) significant lubricant transport becomes evident, as the main portion of the graphite delaminates during sliding and is piled up at both ends of the wear track in a puckered, accordion-like fashion.

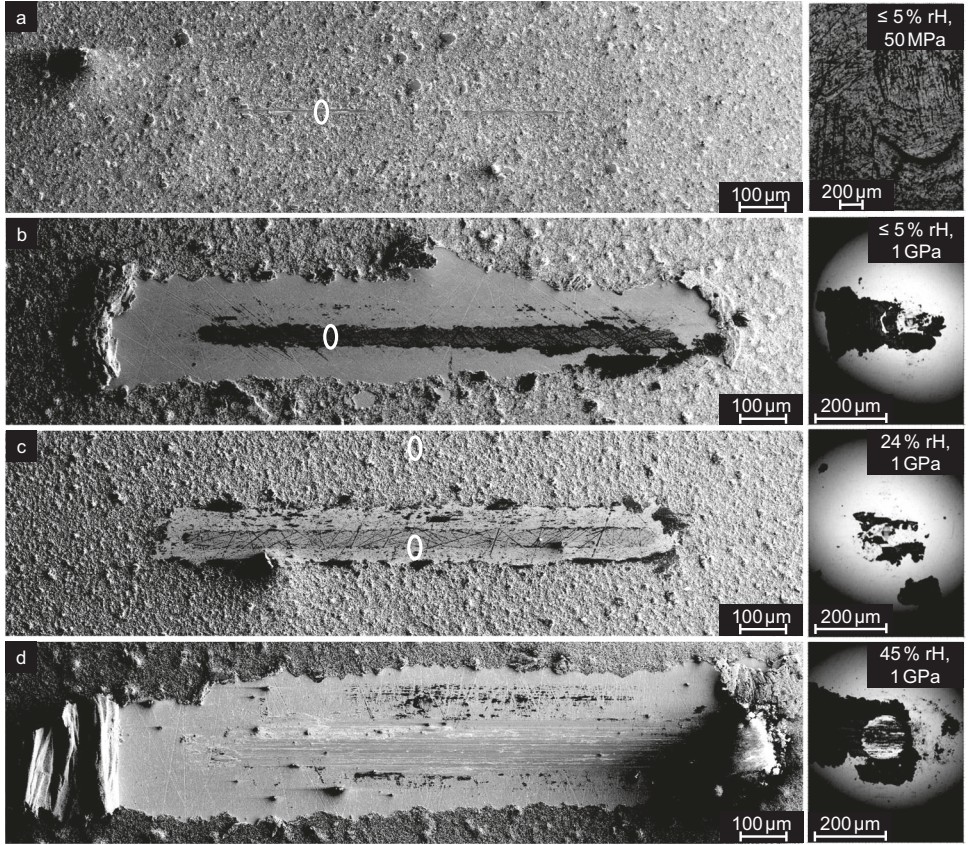

**Fig. 3 | Scanning Electron Microscopy images of wear tracks and counter bodies taken after microtribometer experiments at different pressure and humidity values. a** ≤5% RH (dry pressurized air) at 50 MPa, **b** ≤5% RH (dry pressurized air) at 1 GPa, **c** 24% RH at 1 GPa, and **d** 45% RH at 1 GPa. The white ovals in **a**, **b**, **c** represent regions of interest for a preparation of lamellae that are analyzed in consecutive TEM characterization.

The graphite is transferred onto the counterbody as well. For all the experiments conducted at 1 GPa, a transfer layer was visible on the counterbody (see Fig. 3 right column). Under low load, only little material transfer was observed overall. This material transfer and thus transfer film formation is in coherence with other graphitic sliding experiments reported in literature[18].

To analyze the structure of the carbon layer in the sliding interface in more detail, four 10 μm wide transmission electron microscopy (TEM) lamellae were prepared. One of it from an unworn region as a reference, two after experiments at 1 GPa (≤5% RH and 24% RH) and an additional lamella from the low load sample testet at ≤5% RH and 50 MPa (marked as white ovals in Fig. 3a, b, c). The lamella from Fig. 3b can be found in the SI file, see Supplementary Fig. 2. With the selected three experimental samples the whole range of humidity and normal force is covered. Samples at humidity ≥36% RH were not investigated as the sample was exposed to extensive abrasive wear, thus the carbon layer structure could not be analyzed.

The four lamellae are analyzed by bright field (BF) and high resolution (HR) TEM. The initial micron-thick graphite layer has a porous structure (BF-TEM image in Fig. 4a) resulting from the pile-up of individual graphite flakes during the airbrush deposition process. At larger magnifications, graphitic bands become visible as darker stripes (Fig. 4b) which in turn consist of large bundles with parallel graphene layers (Fig. 4c). Often, these multilayer graphene bundles are terminated by onion-like carbon caps. While the graphene bundles have all kinds of orientations in the bulk of the unworn graphite layer and at the surface, they are always parallel at the interface to the metal substrate (Fig. 4d).

After sliding at $P_{Hertz} = 50$ MPa, the layer is compressed to a thickness of 180 nm (Fig. 4f). The graphitic bands in the bulk of the

coating have survived sliding (Fig. 4f, h), but the porous structure has disappeared and a thin turbostratic structure has formed on top of the coating (see HR-TEM characterization in Fig. 4g).

At $P_{Hertz} = 1$ GPa, the lubricant layer is compressed even more (to a thickness of 30–150 nm; see Fig. 4i, j), and in addition, the graphitic bands have disappeared (Fig. 4j). The graphene bundles in the bulk and at the surface of the graphite layer have vanished and the whole lubricant layer consists of the turbostratic carbon (t-C) structure (Fig. 4k). Thus, the thickness of the t-C is different for low and high load. The parallel graphene bundles remain only at the iron interface (Fig. 4l). In between the graphene bundles and the iron substrate, an additional layer is formed (see area delimited by white dotted lines in Fig. 4l). This layer consists of iron carbides as detected by energy-dispersive X-ray spectroscopy (EDXS) (see Supplementary Fig. 3). The same results were observed for the sample prepared after sliding at 1 GPa and ≤5% RH (see Supplementary Fig. 2), indicating the structural transformation to t-C independently from the surrounding humidity.

Further HR-TEM analysis is performed to elucidate the local structure of the graphite solid lubricant's surface before and after sliding (Fig. 5). Frequently, the unworn graphite is terminated by multilayer graphene bundles. These bundles can be parallel to the graphite surface (Fig. 5a) but are often almost perpendicular to it (Fig. 5b). After sliding, the multilayer graphene bundles have been replaced by t-C structures close to the free surface for both low (Fig. 5c) and high (Fig. 5d) Hertz pressures.

An analysis of the fraction of sp²-hybridized carbon allows further conclusions about the shear-induced phase transition in the graphite layer. Spatially resolved electron energy loss spectra (EELS) of the carbon-K-edge are measured in different region of interest (ROI) (see BF-TEM images in the left column of Fig. 6). The carbon sp² content is

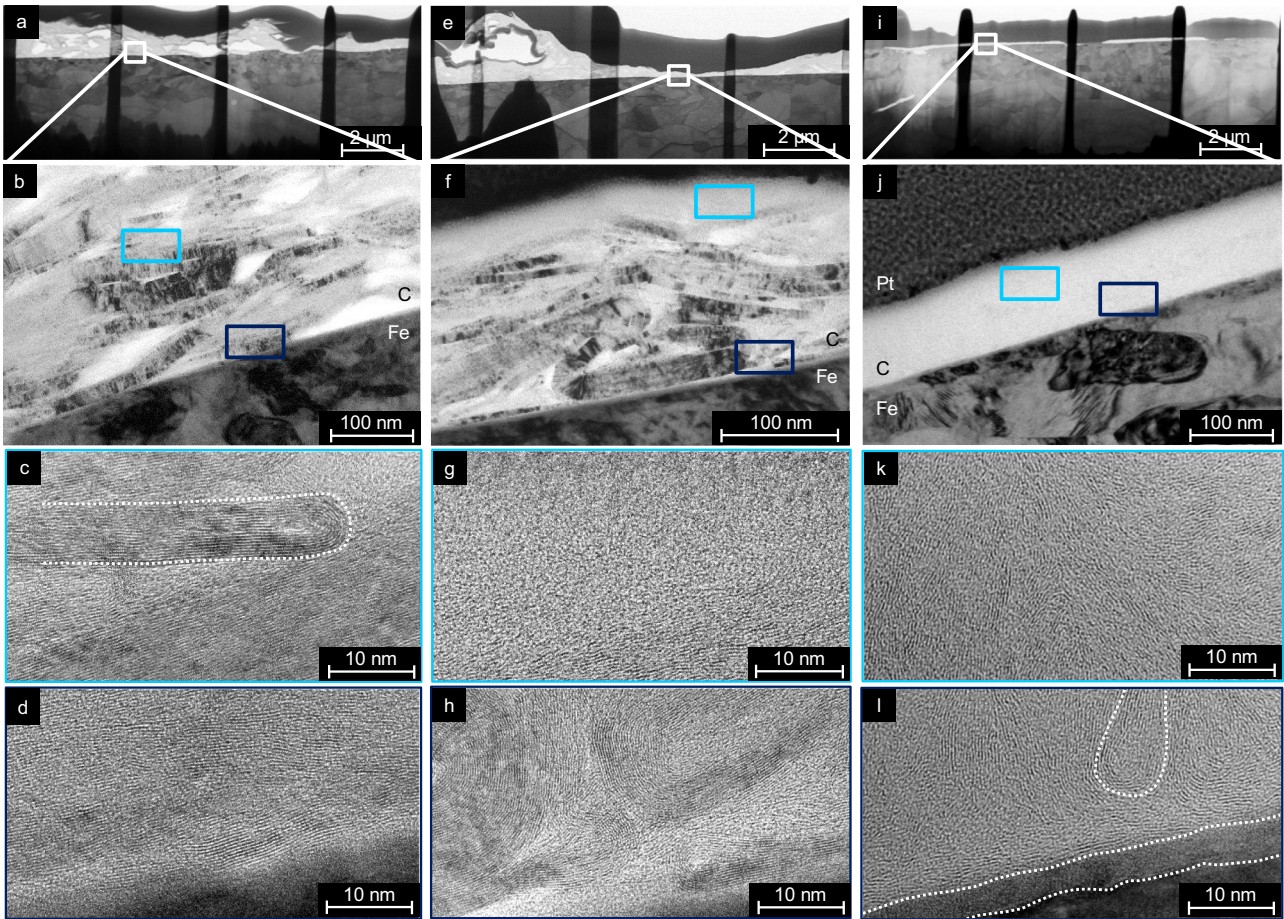

**Fig. 4 | Bright field transmission electron microscopy (BF-TEM, upper half) and high-resolution transmission electron microscopy (HR-TEM, lower half) images.** Taken from TEM lamella prepared in an unworn region (**a**–**d**), after sliding under 50 MPa and in ≤5% RH humidity (**e**–**h**), and after sliding under 1 GPa and in 24% RH humidity (**i**–**l**). Marked in **b**, **f**, and **j** are the iron substrate (Fe), the graphite layer (C), as well as the protective platinum layer (Pt) on the TEM lamella. Marked in light and dark blue are the approximate positions of the HR-TEM studies in the BF-TEM images.

calculated as the ratio of the integrated energy losses of the $\pi^*$ and the $\sigma^*$ state. In the unworn layer, the $sp^2$ content has an average value of $(97 \pm 5)\%$ and is rather homogeneous over the whole measured area (Fig. 6a), hence a purely graphitic coating could be achieved. After sliding at 50 MPa, the $\sigma^*$ peaks in the ROIs closer to the free surface are significantly higher and broader while the $\pi^*$ peaks are diminished (Fig. 6b). This results in a reduction of the $sp^2$ content by more than 20%. A strong gradient in the $sp^2$ content ranging from 73.0% in the highest ROI to 96.5% near the iron substrate can be observed. This correlates well with the partial transformation of the coating to t-C (Fig. 4f).

At $P_{Hertz} = 1\,GPa$, the EELS spectra in the different ROIs are indicating a sizeable reduction of $sp^2$ content (Fig. 6c). Interestingly, the distribution of the $sp^2$-content still exhibits a gradient—with a higher amount of graphite close to the iron substrate (77%) and less at the sliding interface (71%, compare values in the blue rectangles in Fig. 6c).

To conclude the experimental section, the structures found by the HR-TEM prior to and after sliding are schematically presented in Fig. 7. Before sliding, the commercial graphite lubricant of our study consists of a loosely packed array of randomly oriented high aspect-ratio graphite microcrystals (Fig. 7a). These are composed of nanoscale bundles of multilayer graphene—sometimes terminated by onion-like loops, sometimes by sharp edges with individual unconnected graphene. Close to the iron interface a parallel ordering of the bundles is more likely. A homogeneous $sp^2$ hybridisation of almost 100% for the whole coating underlines the high degree of graphitic ordering in the initial lubricant layer.

At low Hertz pressures, sliding leads to the formation of a t-C on top, while the bulk of the graphite coating is slightly compressed but retains most of its original structure (Fig. 7b). Accordingly, the $sp^2$ content develops a pronounced gradient from the Fe substrate to the free interface.

For high pressures, the graphite solid lubricant has completely changed (Fig. 7c). An iron carbide interfacial layer has formed –covered by a thin layer of parallel multilayer graphene bundles. The bulk of the graphite layer has lost its microscale ordering and is transformed into a t-C with feature sizes of the order of 2–10 nm. Most likely the transfer film on the counterbody has experienced the same tribo-induced phase transformation.

In both pressure regimes, the t-C tribo material has increased its $sp^3$ content by more than 20% indicating a significant densification and suggesting that the low friction behavior of this material cannot be explained by a velocity accommodation mode that includes shearing of the t-C bulk. Thus an easy sliding of the t-C (passivated either by aromatic loops or by hydroxylation) against a t-C transfer layer on the counterbody is a more likely explanation of low friction graphite lubrication. Consequently, the deck-of-cards models (Fig. 1a) as well as the adsorption model in its original form (Fig. 1b) can be ruled out for the sliding of commercial graphite lubricants under low and high contact pressures.

## Simulations
To elucidate the mechanisms underlying the shear-induced phase transformation from polycrystalline graphite to t-C, quantum

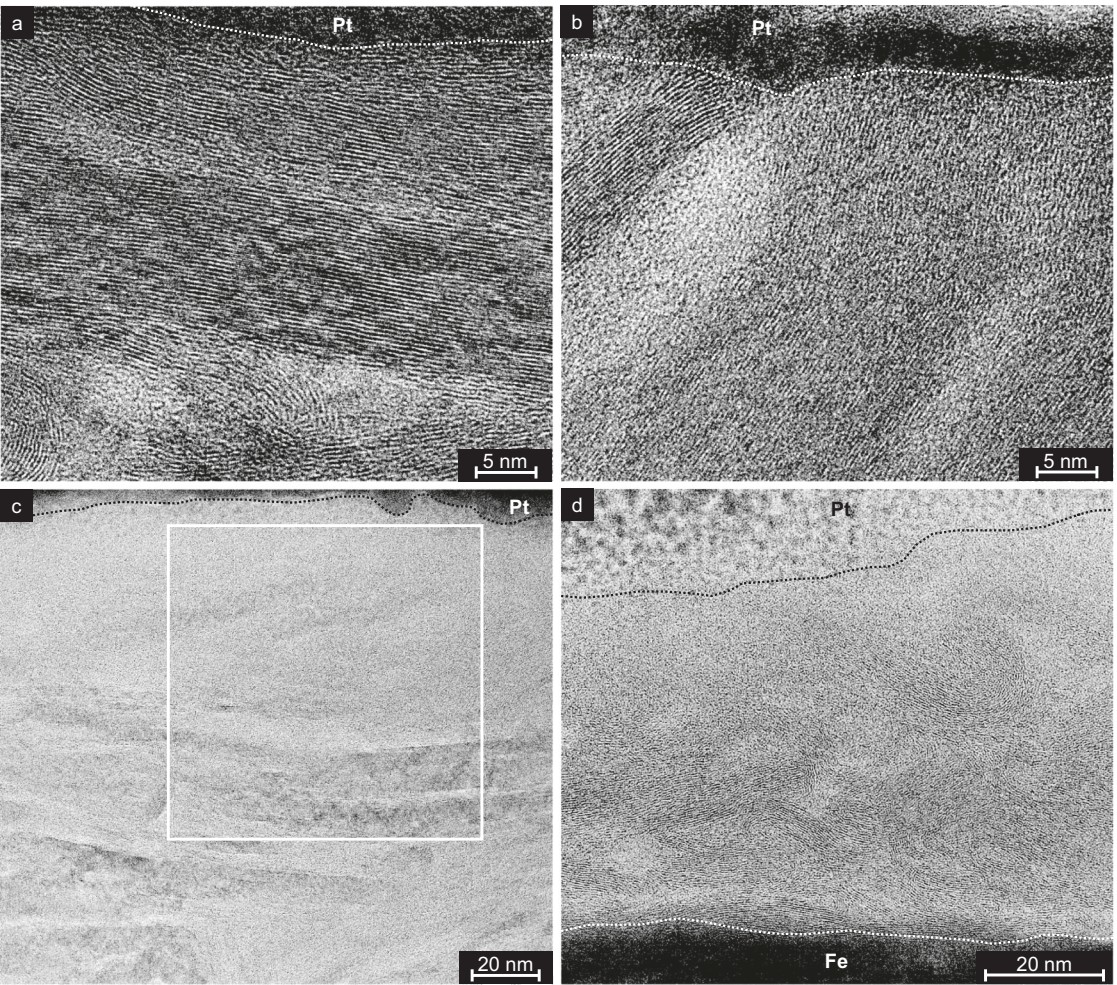

**Fig. 5 | HR-TEM images of the graphite layer at the free surface covered by a Pt layer to avoid damage by TEM preparation.** Images taken prior to (**a**, **b**) and after (**c**, **d**) the sliding experiments at **c** 50 MPa and ≤5% RH; **d** 1 GPa and 24% RH. For a larger magnification of (**c**) see Supplementary Fig. 5, position marked with the square.

molecular dynamics simulations with a DFTB hamiltonian[23] are performed. As a model system for graphite solid lubrication, the mutual sliding of two graphite crystals is considered (Fig. 8). To investigate the effect of humidity, the crystals are separated by $n_{H_2O}$ water molecules prior to sliding. The areal density of water molecules $\rho_{H_2O} = n_{H_2O}/A$ is varied between 0 and 30 nm$^{-2}$. Here, $A$ denotes the lateral area of the sample. The influence of load is studied for normal pressure $P$ ranging from 0.5 to 5 GPa.

Two different orientations of the graphite basal planes (BP) are chosen: BP parallel (Fig. 8a) and BP perpendicular to the sliding interface (Fig. 8c, e, g). The choice of the crystal orientations is motivated by the adsorption model (Fig. 1b) and our experimental TEM analysis of the unworn graphite revealing the frequent occurrence of parallel and perpendicular graphene bundles at the sliding interface (Fig. 5). The crystals with parallel BPs are assumed to consist of ideal graphene sheets without defects and therefore no passivation (for instance by H or OH) is considered. The situation is different for perpendicular BPs. Since TEM images provide no information about the termination of the BPs, we study two different cases: a mixed 50%/50% H/OH passivation and unpassivated terminal carbon atoms.

During sliding, significant differences in tribological behavior emerge depending on crystal orientation, graphene termination, water density $\rho_{H_2O}$, and pressure $P$. First, we focus on the case $P = 1$ GPa and $n_{H_2O} = 16$ corresponding to $\rho_{H_2O} = 14$ nm$^{-2}$. Figure 8a displays a snapshot of two crystals sliding with parallel BPs (see also

Supplementary Movie 1). The evolution of the system's shear stress $\tau(t)$ is depicted in Fig. 8b. Shear stresses are extremely small with an average $\langle \tau \rangle = 0.0008$ GPa. Conversely, sliding of H/OH passivated graphite with perpendicular BPs results in an increased average shear stress $\langle \tau \rangle = 0.16$ GPa (Fig. 8c, d, Supplementary Movie 2). Interestingly, the same shear response is obtained for two unpassivated graphite crystals (Fig. 8e, f, Supplementary Movie 3). Here, it is important to note that after removal of the H/OH terminal groups the individual graphene sheets tend to repassivate by formation of hairpin-shaped loops that connect neighboring graphenes (Fig. 8e). We would like to point out that such loops represent a realistic passivation in experimental samples, since they have been observed in graphite powder after heating to 2000 °C in vacuum[24]. This behavior is also in good agreement with the frequent observation of loops in our TEM analyses and therefore we consider the loop terminated graphite slab as a minimalistic model for chemically unpassivated graphite with perpendicular BPs.

To conclude, already small amounts of water at the sliding interface between two graphite films are sufficient to sustain a good lubricity of graphite. However, what happens upon water starvation at the sliding interface? Let us first consider dry sliding of graphite crystals with perpendicular BPs and complete H/OH passivation (snapshot in Fig. 8g, Supplementary Movie 4). A stick-slip type evolution of the shear stress $\tau(t)$ (Fig. 8h) originates in the interlocking of opposing graphene sheets during sliding. Consequently, the average

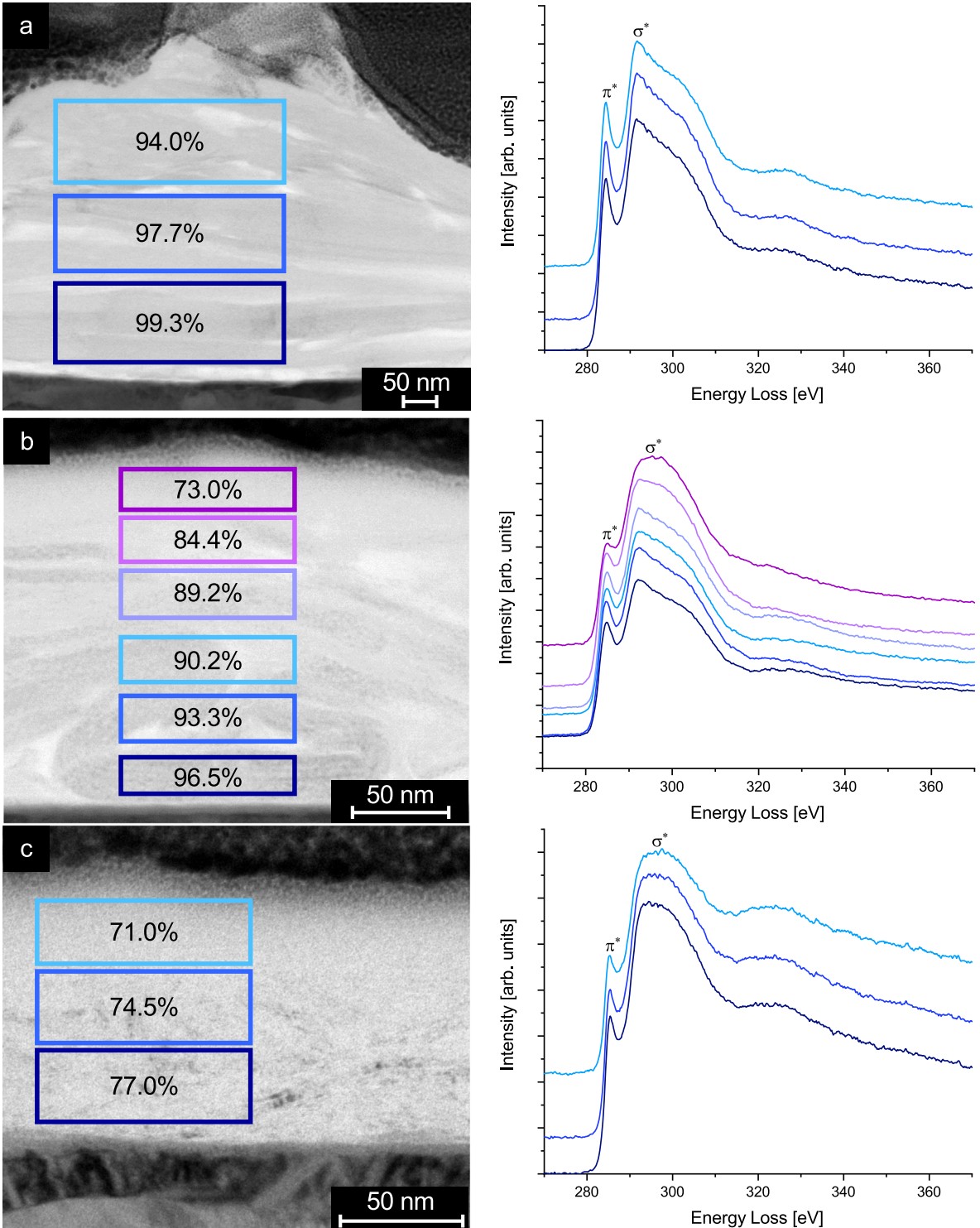

**Fig. 6 | Bright-field transmission electron microscopy images and electron energy loss spectra.** BF-TEM image of the regions of interests (left column) with the corresponding EELS spectra which were taken in a broad, bar-like fashion (right column). **a** Unworn reference layer, **b** graphite after sliding experiment at 50 MPa, **c** graphite after sliding experiment at 1 GPa. The error for the determination of the $sp^2$ content is estimated to be 2%. For the sample at 50 MPa six ROI were taken instead of three, as the TEM analysis exhibited a more changing morphology throughout the whole layer than the other lamellae.

shear stress is increased compared to the water lubricated simulations ($\langle\tau\rangle = 0.26$ GPa).

In order to explore the pressure dependence of $\langle\tau\rangle$, the simulations in Fig. 8a–h are repeated for other normal pressures. A linear relationship between $\langle\tau\rangle$ and $P$ (Fig. 8i) hints towards an Amontonian friction law $\langle\tau\rangle = \mu P$ with $\mu = 0.001$ for water lubricated parallel BPs, $\mu = 0.13$ for both water lubricated perpendicular BP systems and $\mu = 0.24$ for the dry H/OH terminated perpendicular BPs. Even for the highest pressure $P = 5$ GPa, a stable sliding without any structural transitions in the graphite slabs shown in Fig. 8 is established.

Now, we turn to a situation where only one water molecule is inserted between two loop-terminated graphite crystals (Fig. 9a–e, Supplementary Movie 5). After bringing the crystals in contact, immediate cold welding occurs (Fig. 9a). The loops consist of strongly

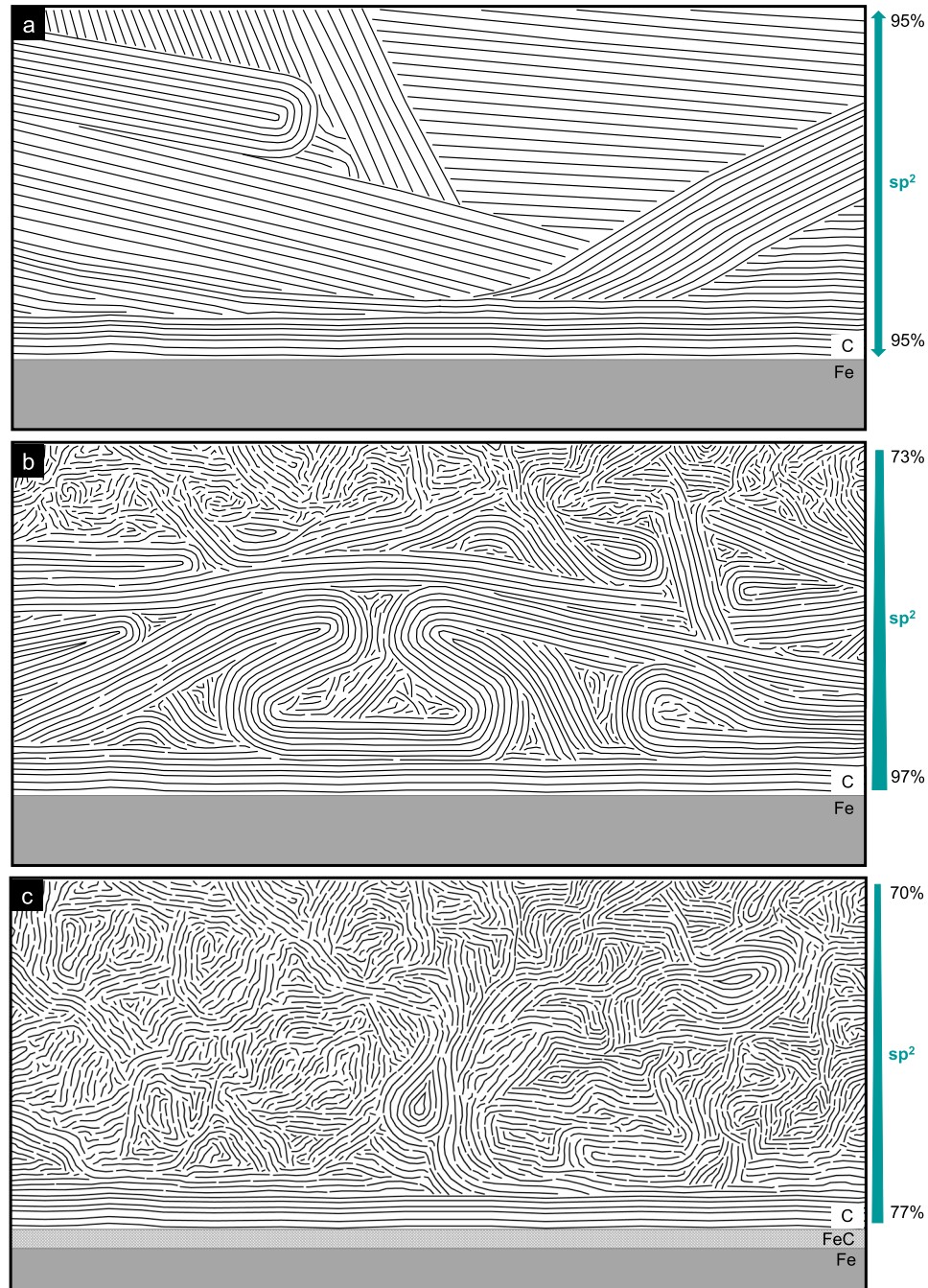

**Fig. 7 | Schematic of the structure in the graphite solid lubricant.** Structure prior to (**a**) and after sliding experiment with low (**b**) and high load (**c**).

curved graphene layers that are very reactive towards out-of-plane chemical bonding. Obviously, one water molecule is insufficient to separate the loops enough to prevent the formation of C−C bonds across the sliding interface (Fig. 9a). During sliding more bonds form leading to the complete opening of the loops and the formation of inter-crystalline graphene bridges (Fig. 9b). This increases the shear stress in the system to values as high as 85 GPa. Further shearing breaks the graphene bridges and generates a t-C structure (Fig. 9c) that develops later-on a sliding interface with onion-like terminations (Fig. 9d). Since the aromatic arcs in Fig. 9d are less curved than the initial loops in Fig. 9a, their reactivity is strongly reduced. Consequently, cold welding is suppressed leading to a drastic decrease of shear stress: $\tau(t)$ in Fig. 9e frequently vanishes for $t > 0.05$ ns. The carbon dimer radical on the upper sliding body in Fig. 9d sometimes

binds covalently to the lower counterbody (Supplementary Movie 5). This transient formation of C−C bonds across the sliding interface results in small spikes in the $\tau(t)$ curve. The water molecule is completely dissociated and hydrogen as well as ether passivation reduce the number of radical sites on the aromatic arcs.

A second simulation of the same system using slightly different starting conditions shows the same initial cold welding and structural evolution (Fig. 9f–j, Supplementary Movie 6). However, the cold welding persists during the complete sliding phase leading to high shear stresses over the whole MD run.

Also a simulation with five water molecules (Fig. 9k–o) first cold welds, but later-on manages to form completely passivated sliding surfaces (see loops, arcs as well as hydrogen, hydroxyl, and ether passivated graphene sheets in Fig. 9n). This lowers $\tau(t)$ to almost zero

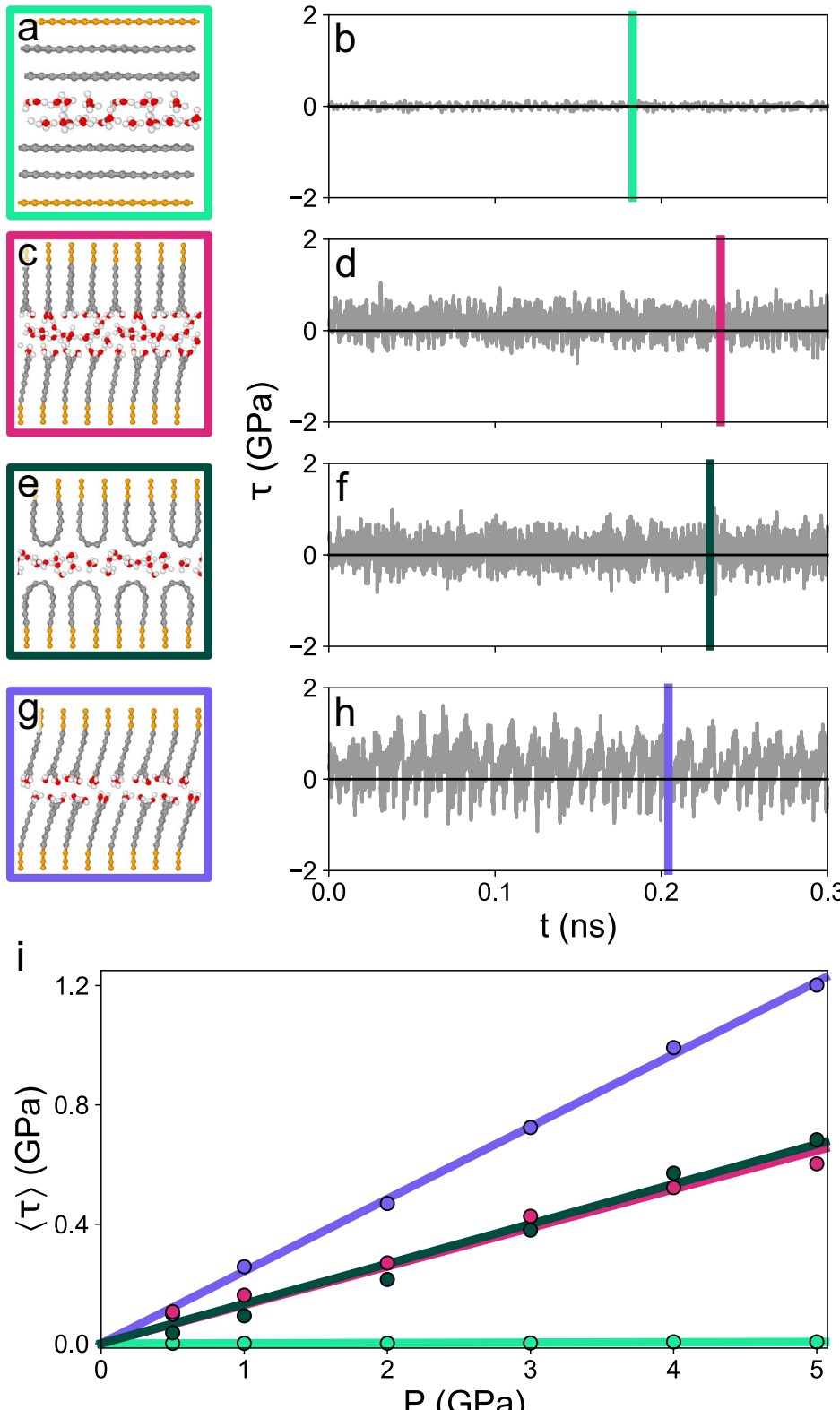

**Fig. 8 | Atomistic simulation of the pressurised sliding ($P$ = 1 GPa) of two graphite slabs.** Basal planes were oriented parallel (**a**, **b**) and perpendicular (**c**–**h**) to the sliding direction. Left column: snapshot of the simulation. Right column: evolution of the shear stress $\tau(t)$. Snapshots are taken at times indicated by vertical bars in the respective $\tau(t)$ curves. **a**, **b** Basal planes of graphite sliding with $n_{H_2O}$ = 16. **c**, **d** Water lubricated sliding of H/OH passivated perpendicular graphite ($n_{H_2O}$ = 16). **e**, **f** Loops formed from initially unpassivated perpendicular graphite sliding with $n_{H_2O}$ = 16. **g**, **h** Dry sliding of mixed H/OH passivated graphite. **i** Average shear stress $\langle\tau\rangle$ as a function of normal pressure $P$ for the systems in **a**–**h**. Color codes refer to the color of the boxes surrounding snapshots. Straight lines are linear fits to the simulation data.

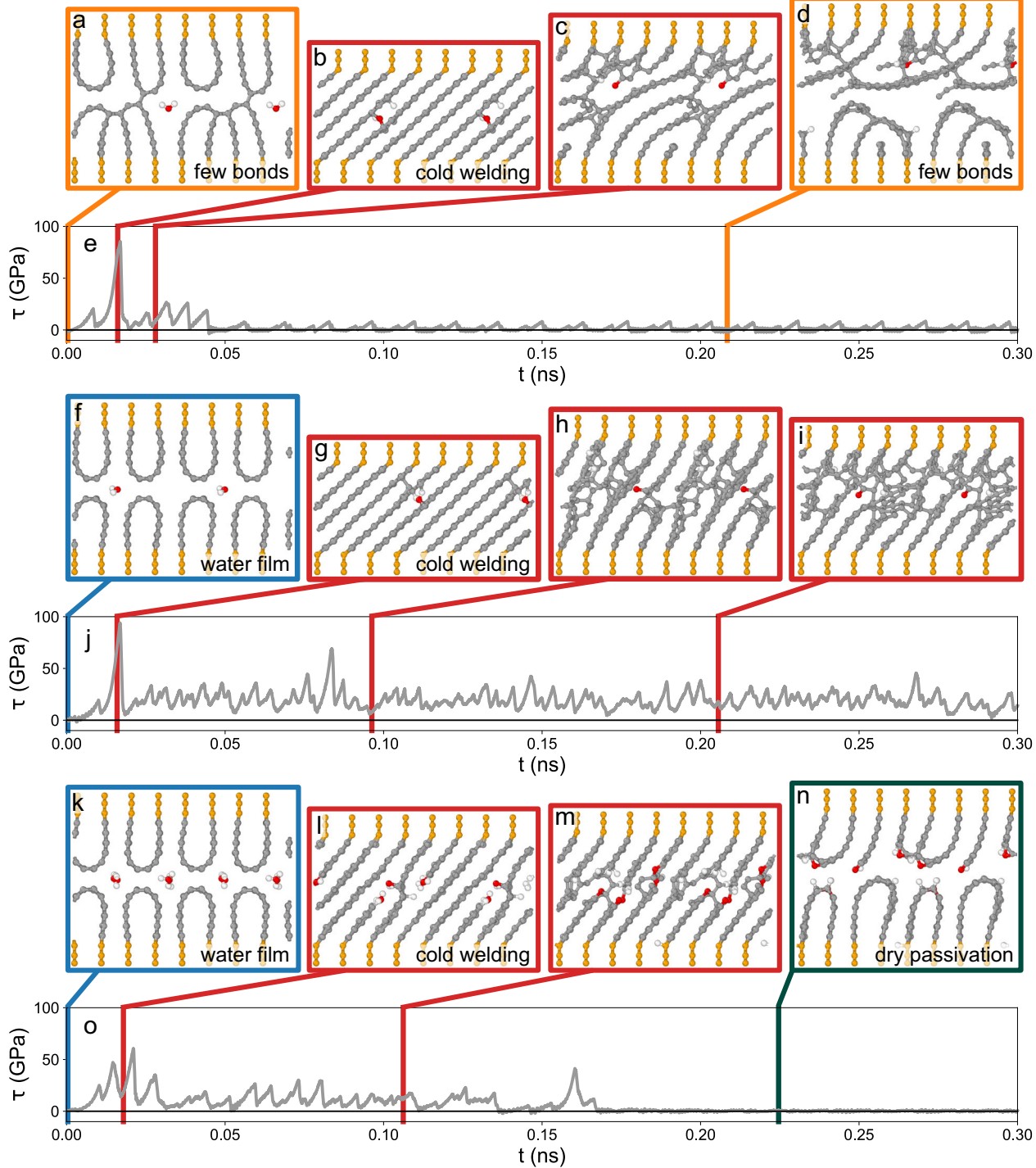

**Fig. 9 | Structural evolution of graphite slabs during sliding in atomistic simulations. a–d** Two loop terminated graphite blocks in contact with one water molecule at a normal pressure of 1 GPa and the corresponding evolution of the shear stress $\tau(t)$ (**e**). Upon sliding, aromatic structures form at the interface. **a–d** Snapshots of the trajectory, duplicated in horizontal direction. **f–i** Snapshots

and shear stress evolution (**j**) of a different trajectory with one water molecule and different initial conditions, during which no surface passivation occurred. **k–o** Snapshots and $\tau(t)$ for sliding with 5 water molecules. After ~0.17 ps the formation of passivated surfaces without free water leads to the separation of the upper and the lower graphite blocks.

for $t > 0.17$ ns (Fig. 9o). Before this transition, shear stresses are already lower than in Fig. 9e, j, since dissociation of the additional water leads to an interruption of graphene bridges (Fig. 9l, m) in the cold welding phase (Supplementary Movie 7).

In order to estimate the probabilities for the occurrence of the different friction states (cold-welded, partially and fully passivated surfaces, and water film lubrication) an extensive MD study of initially

loop-terminated graphite slabs separated with $n_{H_2O} = 0-32$ water molecules is performed. For normal pressures $P = 1$, 3, and 5 GPa several 0.3 ns sliding simulations are conducted and several repetitions with slightly varied initial conditions are carried out. From each trajectory a normalized probability density (PD) of the instantaneous shear stresses is obtained by histograming the $\tau(t)$ curves (Fig. 10a). All trajectories are visually inspected. In combination with the

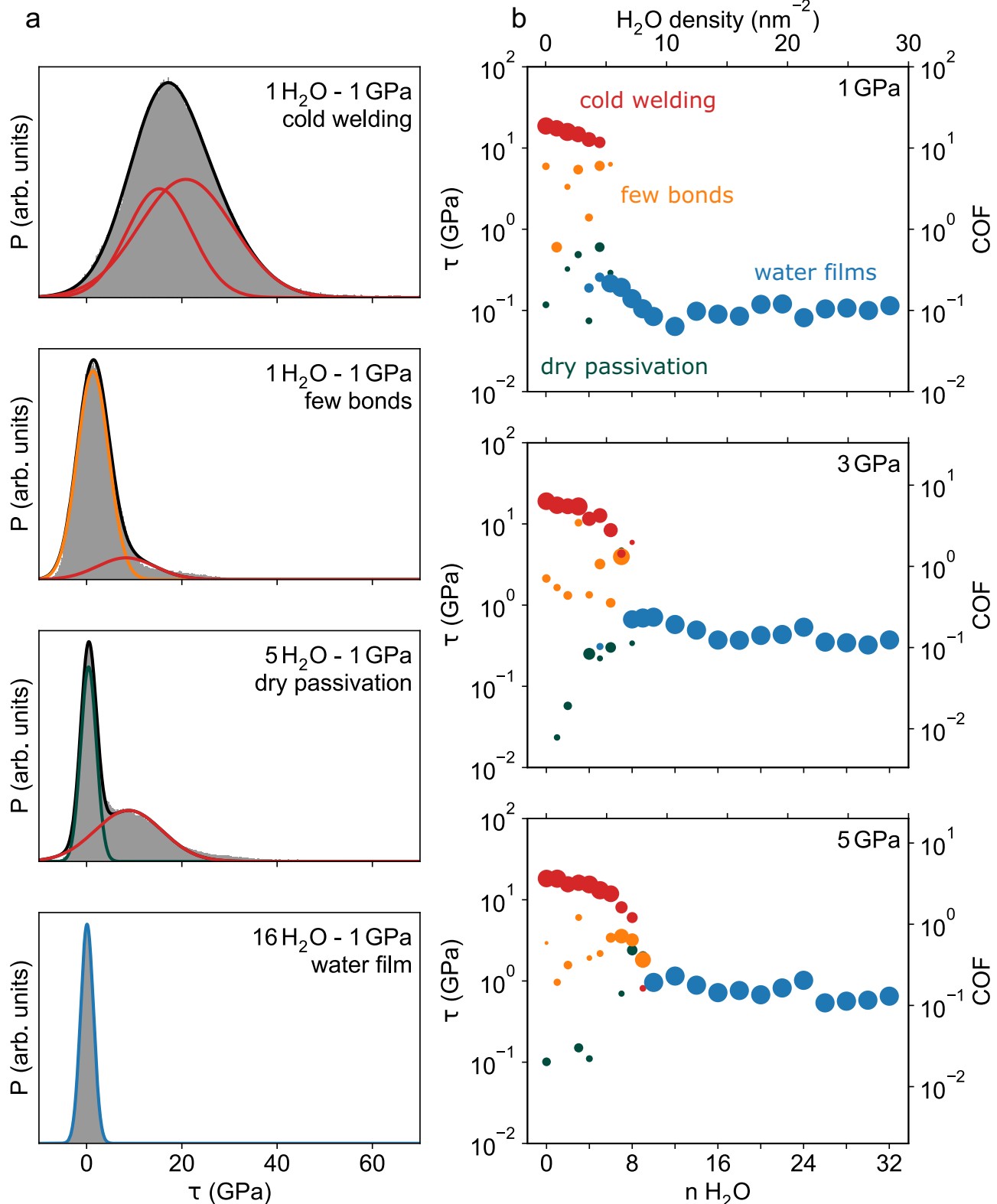

**Fig. 10 | Typical shear stresses in the different regimes. a** Probability densities (P) for the shear stress $\tau$ for the typical system evolution at a normal pressure of 1 GPa. Colored lines show $\chi^2$-fits of gaussians to the probability densities. **b** Average shear stresses determined from the Gaussian fits to the densities as a function of the number of water molecules for normal pressures of 1, 3, and 5 GPa. The colors of the discs distinguish the different regimes, while the disc size indicates the probability of a regime.

characteristic profiles of shear stresses vs. time (Fig. 9), the different states can be identified unambiguously. Since typical shear stresses in the cold-welded, the partially passivated, the full passivated, and the water lubricated states differ significantly, these states occur in the PDs as clearly separated peaks. A set of Gaussian functions is fitted to the PD and associated with the different states based on the described visual inspection. If a transition from one regime to another occurs during the simulation (for example from cold welding to complete passivation; see Fig. 9k–o) a sum of two Gaussian functions is used. Two Gaussians are also required if the cold welded state persists throughout the simulation time (Fig. 9f–j). A typical shear stress for each regime can then be determined from the mean value of the Gaussian(s) for all trajectories with fixed $n_{H_2O}$ and $P$. In the same way the probability to find a system in a certain regime is calculated by an average of the integral over the respective Gaussian(s). The result of this analysis is shown in Fig. 10b.

Depending on the areal water density $\rho_{H_2O}$ in the contact and on the local normal pressure $P$, two major friction regimes emerge in our atomistic simulations. For high water densities, the sliding graphite bodies are separated by continuous water films leading to friction coefficients of the order 0.1 (blue discs in Fig. 10b), while for small $\rho_{H_2O}$ cold welding of the surfaces is observed resulting in high frictional shear stresses of the order of 10 GPa (see red discs in Fig. 10b). The water density $\rho_{H_2O}^c(P)$ marking the transition from high to low friction depends on the local normal pressure in the system. The higher $P$ the more water molecules are required to achieve low friction. For the pressures studied in Fig. 10b $\rho_{H_2O}^c$ ranges between 5 and 10 nm$^{-2}$. Therefore, less than a monolayer of water ($\rho_{H_2O}^{monolayer} \approx 12$ nm$^{-2}$) is already sufficient to enter the water film based low friction regime.

As expected, for $\rho_{H_2O} < \rho_{H_2O}^c(P)$, two additional friction regimes occur. A system can either fully passivate by formation of aromatic arcs, hydrogen, hydroxyl, and ether terminations (Fig. 9n) leading to dry sliding with remarkably low shear stresses (green discs in Fig. 10b) or it achieves only partial passivation resulting in a limited number of interfacial C–C bonds (Fig. 9d) and increased $\tau$ on the order of 1–5 GPa (yellow discs in Fig. 10b).

## Discussion

Our experiments clearly demonstrate that graphite solid lubrication also works at high loads (in the order of GPa), provided the humidity is not too low or too high. Since the measured friction coefficients for sliding at high and low loads are approximately in the same range, it is reasonable to assume that the tribological mechanisms underlying graphite solid lubrication may be identical at high and low loads.

In the following, we discuss these mechanisms and examine the extent to which the results of our work are consistent with the various models presented in the introduction. First, we note that our SEM analysis (Fig. 3) reveals the formation of a thin carbon film on the ball sliding against the plate covered by the graphite coating (in the case of low loads) or a thick carbon layer (in the case of high loads). In consequence, sliding takes place between the carbon layer on the counterbody and the carbon layer formed on the iron plate for the majority of the experimental duration. This observation underlines the importance of transfer film formation in graphite solid lubrication and also justifies the use of two graphite plates as starting configurations in atomistic sliding simulations. Before discussing our experimental results further, three main conclusions should be drawn from the DFTB-MD simulations performed in this article.

First, the areal density of water molecules $\rho_{H_2O}$ represents an important control parameter that determines the major friction regime in an asperity collision between the graphite lubricant and a transfer film on the tribological counterbody. Less than a monolayer of water suffices to protect the contacting carbon films from cold-welding. The water plays a dual role in this context: as a source of passivating chemical species (H, OH, and ether) and as inert spacer molecules that separate the surfaces and attenuate the formation of C–C bonds at the interface.

Second, if contacts between two graphite crystals run dry (i.e., $\rho_{H_2O} < \rho_{H_2O}^c$) cold welding is likely to occur if the passivation is not strong enough. This happens when the basal planes are not parallel to the slip direction and end in tight bends, as the carbon atoms in highly curved graphene structures easily change their hybridisation from sp$^2$ to sp$^3$. It is likely that incomplete chemical passivation of the graphene edges also contributes to cold welding.

Third, cold-welding represents the crucial trigger that drives the tribo-induced phase transformation of insufficiently passivated graphite crystals into a more inert turbostratic carbon tribomaterial. The fact that t-C is observed in our experiments at low and high normal loads indicate that the condition $\rho_{H_2O} < \rho_{H_2O}^c$ is fulfilled for many asperity collisions (at least during running-in) irrespective of $P_{Hertz}$. However, as seen in Fig. 10b the probability for amophization increases with an increase in load. At higher load (e.g., 1 GPa), the density of sufficiently high local pressures to promote water squeeze out increases, allowing the formation of t-C in a larger depth of the coating.

We would like to point out the difficulty in establishing a direct relationship between the relative humidity in our microscale tribometer experiments and the thickness of water films in our nanoscale sliding simulations. Experiments suggest that closed water films are present on HOPG surfaces at relative humidity as low as 10%[25]. In Grand Canonical Monte Carlo simulations of an unloaded nanoscale ta-C tip sliding on HOPG several monolayers of water were observed in the nanocontact for RH > 40%[26]. In principle, corresponding simulations could be performed for our graphite lubricant to relate RH with the water film thickness in our experimental microscale contacts. However, the size of the contact area renders such simulations extremely challenging. This would exceed the scope of this article and thus, we are limited to a qualitative discussion. In any case, it can be safely concluded that high pressures in combination with surface roughness and low sliding speeds result in submonolayer water films that promote cold welding and t-C formation.

Usually, ambient humidity leads to rapid condensation of water films on the graphite outside the tribological contact acting as a water reservoir. Therefore, low friction in graphite solid lubrication requires moist conditions that support water transport into the tribocontacts. Conversely, high friction is expected when many asperity contacts run dry. For instance, a massive drop in humidity can dry out the outside reservoirs leading to a subsequent break down of water monolayer lubrication. This argument was used by Savage[9] in his adsorption model to explain the frequently observed spontaneous onset of severe wear on graphite brushes in aircraft generators at high altitudes [27].

As the experiments have shown, under moist conditions below 37% RH the carbon material remains in the wear track as a thin carbon layer, lowering the steady state friction coefficient to 0.10–0.14 which is well in the range predicted by our simulations. The original adsorption model assumes that this carbon material retains its graphitic character. Indeed, our TEM analysis show that the initial graphite coating consists of loosely packed individual graphite flakes with a porous structure. EELS analysis confirm the graphitic nature of the initial coating with a measured sp$^2$ content of 97%. During sliding however, the porous structure experiences strong compression and shear forces that trigger a tribo-induced phase transition. This transition is visible in the measured sp$^2$ content reduction in Fig. 6 and in additional Fast Fourier Transformation (FFT) analyses of the HR-TEM images, see Supplementary Fig. 6. In contrast to the adsorption model, no ordered graphite is observed at the sliding interface; instead a t-C layer forms that increases its thickness with an increase in normal load. Consequently, we must conclude that although our investigations support many of the conclusions of Savage's pioneering work, the adsorption model is incomplete with respect to the formation of carbon tribomaterial. Therefore, we suggest an extended adsorption model as sketched in

Fig. 1d, which is valid for a broad range of loads and humidities (as evident from the data collected at 50 MPa/RH < 5% as well as 1 GPa/RH < 5%, for the latter see Supplementary Fig. 2). This extension is crucial since the formation of t-C is responsible for the observed long lifetime of the coating and allows for a large reduction of friction.

The transformation of graphite to t-C has been reported previously[28–30] but only during milling of graphite, where more severe conditions prevail. Thus, our research shows for the first time that sliding triggers t-C formation. Recently, the tribological performance of multilayer graphene[31] or t-C[32] in high pressure experiments and ambient conditions has been investigated. Kumar et al. observed similar trends in terms of decreasing friction with increasing humidity, but measured higher overall friction. Bhowmick et al. investigated multi-layer graphene[31]. These authors found similar trends regarding the formation of amorphous carbon, although the amorphization took only partially place producing graphene stacks embedded into the amorphous material without a height gradient in the $sp^2$ hybridization. In our work, a $sp^2$ gradient was evident both from HR-TEM images as well as quantitative EELS analyses. The graphene roller mechanism as proposed by Bollmann and Spreadborough[13,33] could be omitted as no rollers were observed during the MD simulations or in the HR-TEM analyses; if applicable to our experiments, we would have observed rollers at least on the sides of the wear tracks.

But what happens under extremely moist conditions? In our experiments with high humidity (≥37% RH) the steady-state low friction regime was only short-lived and followed by an abrupt rise in friction. This observation is accompanied by extensive material transport of the graphite coating out of the wear track, resulting in a breakdown of solid lubrication and CoF values that are comparable to those of bare steel-steel contacts as shown by previous studies[22]. This trend is also visible in the wear behavior in Fig. 2d and the wear scar in Fig. 3d indicating massive cold-welding despite the presence of abundant water in the system. As a working hypothesis, an excess of water on the graphite lubricant could promote dissolution of the graphite crystals or even a delamination of the coating. Above a certain threshold (here ≥45% RH) water might diffuse into and especially underneath the graphite coating in larger quantities, hence forming internal water films. As a consequence, the lubricant would be removed during the first sliding cycles, as observed in our experiment. To corroborate this scenario, additional microtribometer experiments are conducted where the sphere was immersed into a drop of water placed on the graphite-coated plate; hence the effect of capillary condensation can be ruled out. As evident from Fig. 2b the CoF immediately rises to high values after a few cycles and exhibited high wear (Fig. 2d) which hints to a quick removal of the coating. As a further test of our dissolution scenario, a graphite-coated plate was completely immersed into deionized water and put into an ultrasonication bath. Already after a few seconds delamination of the coating is observed resulting in its complete removal after 1 min, as evident from Supplementary Fig. 4. This suggests that the observed increased wear at high relative humidity is a consequence of the dissolution of the coating by water.

In conclusion, we explored in this article the suitability of a technical graphite coating for solid lubrication applications at high loads (for example in rolling bearings). Although various analyses of our experimental wear scars combined with quantum MD calculations indicate that the basic reasoning of the adsorption model is also valid for the explanation of graphite lubrication at high loads, our work suggests that t-C plays an additional important role. An adsorption model extended by a t-C tribolayer allows us to close a gap in the understanding of graphite lubrication.

## Methods
### Sample preparation
The tribological experiments were conducted with iron plates (Goodfellow GmbH, Germany, purity 99,5%, 0,9 mm, as milled). The polishing was done with a semi-automatic grinding machine (Buehler PowerPro 4000, Buehler Ltd, Germany) after which the plates exhibited a surface roughness $S_a$ of (81 ± 4) nm. In an ultrasonication bath, the samples were afterwards cleaned for 10 min in succession with acetone and isopropanol. Drying with pressurized air was conducted to avoid corrosion.

For the graphite coating a suspension (L-GP 386 ACHESON, Henkel, the Netherlands) was applied with an airbrush spray gun (Harder & Steenbeck Ultra Solo Double Action, Harder & Steenbeck GmbH & Co.KG, Germany). The airbrush was moved manually in 30 consecutive sweeps across the substrate surface at a distance of 15 cm. A consistent thickness of 3.5 μm was validated for all plates after the coating process by confocal microscopy. To ensure complete evaporation of any solvents, the coated plates were left in ambient conditions for two days prior to any experiment.

### Microtribometer experiments
The graphite covered iron plates were tested in linear reciprocating microtribometer (Tetra Basalt Must, Tetra Ilmenau, Germany) in a sphere-on-flat setup, as described in detail in[34]. Two counterbody materials were used: a 100Cr6 steel sphere (G10, $r$ = 1 mm, Spherotech GmbH, Germany) for the experiments conducted under high load and a spherical steel cylinder ($r$ = 11 mm, DIN 6325, Wegertseder) for the low load experiments. In both cases the material was glued with cyanoacrylate glue onto a double-leaf cantilever with spring constants of $k_t$ = 4.861 and $k_N$ = 1.903 mN μm⁻¹. For the removal of potential glue residuals the counterbody was cleaned with acetone and isopropanol prior to the experiment. To ensure reproducibility, all experiments were conducted with different batches of the materials, on different days, and repeated at least three times (the low load experiment was only repeated once due to the increased complexity.). Each experiment was conducted in an environment of ~30 °C temperature and with a new counterbody for each experiment.

A normal force of 402 mN or 5,6 mN was applied onto the counterbody, resulting in a Hertzian contact pressure of 1008 MPa or 50 MPa, respectively. For 500 linear reciprocating cycles, a stroke length of 1 mm and a speed of 0.5 mm s⁻¹ was maintained. To analyze the humidity influence on friction and wear of graphite lubrication, the humidity in the experimental chamber was adjusted with a self-build humidity controller based on an ultrasonic atomizer. To validate the present humidity, a hygroscope was used (LOG210 PDF-data logger, DOSTMANN electronic). At low humidity (0–20% RH) the hygroscope has a uncertainty of ±5%[35]. An estimation of the flash temperature as well as the capillary force was conducted and can be found in the SI file. Increase of temperature due to sliding as well as normal forces due to capillarity is negligible.

### Analysis
The topography, as well as ex situ wear volume, of all wear tracks and counterbodies was analyzed by means of confocal microscopy (Sensofar Plµ 2300, Sensofar, Spain). To calculate the wear volume $V$ of the plate, three sections at 0.3, 0.5, and 0.7 mm distance from the start of the wear track and perpendicular to it were chosen. Their average was thus multiplied with the length of the wear track.

The counterbody was wiped with acetone prior to wear measurements to wipe away particles, followed by the determination of the wear calotte width $D$ by confocal microscopy. Together with the radius $r$ of the sphere, the height $h$ of the worn spherical cap can be calculated (see Eq. (1)). The wear volume $V$ of the sphere can thus be calculated according to Eq. (2)[36]. To yield the wear coefficient $k$ for both counterbody and plate, the wear volume $V$ is divided by the applied load $L$ and the overall sliding distance $d$ (in these experiments 1 m), see Eq. (3). The wear data for the counterbodies can be

found in Supplementary Fig. 1.

$$h = r - \sqrt{r^2 - \frac{D^2}{4}} \qquad (1)$$

$$V = \left(\frac{\pi h}{6}\right)\left(\frac{3D^2}{4} + h^2\right) \qquad (2)$$

$$k = \frac{V}{L\,d} \qquad (3)$$

Scanning electron microscopy images (SEM) were taken of the individual wear tracks (Helios NanoLab DualBeam 650, ThermoFisher Scientific, USA).

## TEM analyses

Detailed TEM analyses were conducted on four lamellae. Three were prepared from the sample prior and after the 1 GPa experiment, one each perpendicular to the middle of the wear track after the experiment at 24% RH and ≤5% RH (for the latter see Supplementary Fig. 2), and one from an unworn, reference region. A forth lamella was taken from the 50 MPa sample in the wear track after the experiment. Structural and microchemical investigations of the graphite layers deposited on steel were performed by conventional TEM, HR-TEM, EDXS, as well as EELS. For this purpose, an electron-transparent cross-section specimen was prepared by focused ion beam (FIB) milling using an FEI Strata 400 dual-beam instrument. Before FIB preparation of the TEM lamella, a Pt protection layer was deposited on the original sample surface. Coarse FIB milling was done at 30 kV accelerating voltage with Ga+ ions, whereas for final polishing of the sample surfaces (ca. 5000 scans at each side) 5 kV with -70 pA ion current was used. TEM investigations of the graphite layer were carried out with an FEI Titan3 80-300 (Thermo Fisher Scientific, Waltham, USA) equipped with a Gatan imaging filter of the type Tridiem 865 ER (Gatan Inc., Pleasanton, USA). For EELS analyses, the Titan microscope was operated in the scanning TEM (STEM) mode at 300 kV, where during spectrum recording the electron probe was scanned across an area of about $20 \times 20$ nm in size in order to reduce possible changes of the chemical bonding of carbon atoms. To record the carbon K-edge (ionization energy of 284 eV) spectra were taken in the range from about 200–610 eV with a dispersion of 0.2 eV per channel. According to the full width at half maximum (FWHM) of the zero-loss peak, the energy resolution was -0.7 eV. A typical measuring time for the acquisition of C-K edge amounted to 1–3 s, and several spectra were accumulated for each selected sample area. For spectrum processing, which includes background subtraction and deconvolution to correct for multiple scattering, the Gatan program Digital Micrograph was used.

## Simulations

Based on our experimental observations, we primarily consider systems in which the graphene layers are oriented perpendicular to the sliding direction. Our models consist of two graphite crystals with cross-sectional areas of $13.42 \times 8.52$ Å$^2$. The $\langle 10\bar{1}0 \rangle$ direction of the graphite crystal is oriented perpendicular to the contact area, while the $\langle 0001 \rangle$ direction is oriented along the sliding direction. The systems contain 256 carbon atoms, periodic boundary conditions are applied in all directions and dangling bonds at the top- and bottommost carbon atoms of the graphene layers are terminated with hydrogen atoms. We would like to point out that our model systems do not allow a direct comparison with the experimental friction behavior. The high computational cost limits DFTB-MD simulations to system sizes of about 1 nm in each spatial direction and simulation times of <1 ns.

Typical sliding velocities in the simulation are five orders of magnitude higher than in the experiments (100 m s$^{-1}$ vs. 0.5 mm s$^{-1}$). Therefore, the frequency of thermal barrier crossings is underestimated and most reactions occur via stress-induced instabilities. However, it is evident from previous studies that simulations at higher shear rates still provide useful qualitative insights into mechanochemical reactions in carbon based tribosystems[37,38].

For the surfaces facing the sliding interface, we consider two different terminations of the graphite crystals. First, unterminated, ideal graphene edges (Fig. 8e), and second, a complete passivation with H/OH groups. In the latter case, we randomly choose which carbon atoms are passivated with hydrogen and which with hydroxy groups (Fig. 8a, b).

To model the influence of humidity, between 1 and 32 water molecules are introduced into the region between the crystals. The contact pressure is controlled using a damped barostat coupled to two stiffly held zones at the top and at the bottom of the system[39] (orange atoms in the snapshots in Fig. 8a, c, e, g). To gain information about the probabilities of occurrence of the different observations, all simulations with up to nine water molecules are repeated ten times with randomly varied initial positions of the water molecules, for systems with 10–16 molecules four repetitions are performed, systems containing 18 and more water molecules are carried out only once.

We use density functional tight-binding (DFTB) molecular dynamics simulations[23] to study the behavior of our model systems under tribological load. The use of DFTB ensures accurate modeling of chemical reactions between water molecules and graphite surfaces. A Langevin thermostat[40] at 300 K acting along the $\langle 1\bar{2}10 \rangle$-direction of the graphite lattice only, i.e., perpendicular to the sliding direction and perpendicular to the graphite surfaces, controls the temperature of the system. This ensures efficient temperature control and simultaneously minimizes the influence of the thermostat on the interactions between the water molecules and the graphite surfaces. The equations of motion are integrated using the velocity Verlet algorithm[40] with a timestep of 0.1 fs. Prior to sliding, the atomic positions are relaxed and the crystals are pressed against each other until equilibrium is reached. Shear is then applied by moving the upper stiff layer with a constant speed of 100 m s$^{-1}$ over a period of 300 ps.

In Figs. 9 and 10 of the main text, simulations with identically oriented loop-terminated graphite crystals are shown. In experimental contacts, a wide variety of crystal orientations will occur. Thus, our parallel loop simulations represent the limiting case of well-aligned surface terminations. We report results for another limiting case in the Supplementary Information (Supplementary Fig. 7). Additional simulations with the upper crystals rotated by 90° around the normal of the sliding plane (i.e., loops on the upper crystal are perpendicular to loops on the lower one) exhibit a similar behavior. Also here, two main regimes are found. Already a water submonolayer is sufficient for water film lubrication, while water starvation causes cold welding. Interestingly, reformation of aromatic structures is not observed within the timescale of these simulations indicating that alignment of two cold welded graphite crystals favors t-C formation.

## Data availability

The initial atomic configurations of the DFTB-MD simulations have been deposited in the Zenodo repository at https://doi.org/10.5281/zenodo.7037629. The experimental raw data generated in this study have been deposited in the KITOpen repository openly accessible under https://doi.org/10.5445/IR/1000142811.

## Code availability

The DFTB simulations were performed using the open source code ATOMISTICA[41] and the mio-1-1 parameter set[23,42]. Trajectories were visualized with the open source software OVITO[43].

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

## Acknowledgements

This work is funded by the Deutsche Forschungsgmeinschaft (DFG). The project "Mechanism of Graphite Lubrication in Rolling Contact" ("Mechanismen der Graphitschmierung in Wälzkontakten") is part of the priority program SPP2074 (grant number DI1494/7-1 (C.E.M. and M.D.) and MO879/20-1 (A.K. and M.M.). We thank Reinhard Schneider and Erich Müller from the Laboratory for Electron Microscopy (LEM KIT, Karlsruhe) for performing the transmission electron microscopy measurements. Thanks also to Harun Candan and Radoslav Yankov for conducting the microtribometer experiments. We gratefully acknowlege useful discussions with Andreas Kailer. Atomistic simulations have been carried out on the JUWELS cluster at Forschungszentrum Jülich (grant HFR19, A.K. and M.M.).

## Author contributions

C.E.M. and A.K. contributed equally to this work. C.E.M. performed the experimental studies and analyses. A.K. performed the simulation studies and analyses. C.E.M. and A.K. visualized the data. M.D. and M.M. supervised the work. All authors worked on the conceptualization of this work as well as the drafting, writing, and reviewing of the paper.

## Funding

## Competing interests

The authors declare no competing interests.
