## [Peer Review File · Nature Communications]

Title: Humidity-dependent lubrication of highly loaded contacts by graphite and a structural transition to turbostratic carbonREVIEWER COMMENTS

Reviewer #1 (Remarks to the Author):

It has been known for several decades that graphite friction and wear decrease significantly at high relative humidity. At low loads, this has been attributed (amongst other potential mechanisms) to the formation of water layers adsorbed on top of the graphite, which act as boundary lubrication films (adsorption model). Here, the authors investigate using microtribometry and quantum molecular dynamics simulations whether this mechanism also prevails at high loads, which are relevant to many machine components.

The authors confirm that the high to low friction transition with increasing relative humidity also occurs for graphite at high loads. Transmission electron microscopy (TEM) shows a transition of the polycrystalline graphite lubricant into turbostratic carbon (t-C) after high (1 GPa) and even after low load (50 MPa) sliding. Quantum (DFTB) molecular dynamics simulations relate high friction and wear to cold welding and shear-induced formation of t-C. Low friction is maintained when molecular water films are preserved, even on t-C surfaces.

The manuscript is well-written, the methods are robust, and the results should be of interest to the tribology community. However, I am not sure whether it is of sufficient general interest for this journal given that the results are not particularly surprising. The results effectively confirm that the previously proposed adsorption model is applicable over a wide load range, but that stress-induced transformation of graphite to t-C at the sliding interface should also be accounted for. Some specific comments and recommendations are given below.

1. Introduction: other proposed mechanisms of graphite lubrication, e.g. roller mechanism,[1,2] could also be discussed here. Indeed, this mechanism could be related to the transformation of graphite to t-C.
2. Introduction: The schematics could be made more visually appealing, particularly Figure 1. For example, a 3D rendering showing water molecules, fragments, surfaces etc.
3. Results: It should be noted here that t-C is commonly formed from the mechanical milling of graphite,[3] which also applies sliding forces (shear stresses). It is therefore perhaps unsurprising that this also occurs inside a tribological contact.
4. Results: While the TEM images provide some evidence of transformation to t-C, further surface analysis, e.g. Raman spectroscopy or XRD could be used to confirm this. This is important since it is one of the central claims of the paper.
5. Results: How are the selected water coverages justified compared to experiments? See also comment 8.

6. Discussion: Other studies that have compared the tribology of graphite to t-C coatings could also be mentioned here.[4-6]

7. Discussion: Very similar observations to those made in the current study (although with slightly different interpretations) have recently been made using multilayer graphene rather than graphite as a lubricant (on a different metal substrate).[7] This previous study should be discussed here.

8. Discussion: A more detailed explanation of the increase in friction and wear at very high relative humidity (capillary condensation) should be provided. Generally speaking, capillary condensation only has a significant effect on friction for hydrophilic surfaces.[8] Since graphite is hydrophobic, this supports the authors claim that the interface is passivated by polar groups that increase its hydrophilicity. Most measurements of the hydrophobic effect are from nanoscale single asperity studies using atomic microscopy. However, the authors should attempt to quantify the potential contribution of capillary effects to adhesion and friction in this system. Some methods to estimate the contribution of capillary condensation to friction in multi-asperity contacts are given in Ref. [9].

9. Methods: In the DFTB calculations, the number of water molecules is varied between 1 and 32, which is supposed to represent different relative humidity values in the experiments. In previous classical MD simulations of graphite friction in the presence of water, Grand Canonical Monte Carlo calculations have been used to explicitly link relative humidity to graphite surface coverage.[10] While I recognise that GCMC calculations would be prohibitively expensive using DFTB calculations, performing such calculations for the current systems using classical MD would enable a more robust link between the relative humidity in the experiments and the number of water molecules in the simulations. Note that given the very low pressure-viscosity coefficient of water, fluid entrainment (and its load dependence) will be negligible for the current systems. Thus, elastohydrodynamic lubrication analysis (as suggested as a potential solution in the Discussion section) is unlikely to give an accurate estimate of the water coverage at the sliding interface.

10. Methods: Are the tribology experiments expected to be isothermal? If not, please provide an estimate of the amount of frictional heating to be provided (using Archard equation [11] or similar).

11. Methods: As is common when comparing molecular simulations to experiments, the sliding velocity in the DFTB simulations (100 m/s) is much faster than in the associated experiments (0.5 mm/s). The friction behaviour cannot reasonably be expected to be independent of sliding velocity over this range for the studied systems and this should be acknowledged.

12. Methods: Related to the point above, thermostating the entire system to 300 K prevents thermal gradients from forming, as they do in real experiments.[12] Moreover, thermostating confined fluid molecules (water here) can artificially affect their behaviour, particularly when using a stochastic thermostat such as Langevin. Thus, it is usually preferable to thermostat only the outer layers of confined systems.

References

1. Bollmann, W. & Spreadborough, J. Action of Graphite as a Lubricant. *Nature* 186, 29–30 (1960).
2. Spreadborough, J. The frictional behaviour of graphite. *Wear* 5, 18–30 (1962).
3. Li, Z. Q., Lu, C. J., Xia, Z. P., Zhou, Y. & Luo, Z. X-ray diffraction patterns of graphite and turbostratic carbon. *Carbon*. 45, 1686–1695 (2007).
4. Kumar, N., Dash, S., Tyagi, A. K. & Raj, B. Super low to high friction of turbostratic graphite under various atmospheric test conditions. *Tribology International* 44, 1969–1978 (2011).
5. Kumar, N. et al. High-temperature phase transformation and low friction behaviour in highly disordered turbostratic graphite. *Journal of Physics D: Applied Physics* 46, (2013).
6. Kumar, N., Kozakov, A. T., Ravindran, T. R., Dash, S. & Tyagi, A. K. Load dependent friction coefficient of crystalline graphite and anomalous behavior of wear dimension. *Tribology International* 88, 280–289 (2015).
7. Bhowmick, S., Banerji, A. & Alpas, A. T. Role of humidity in reducing sliding friction of multilayered graphene. *Carbon* 87, 374–384 (2015).
8. Binggeli, M. & Mate, C. M. Influence of capillary condensation of water on nanotribology studied by force microscopy. *Applied Physics Letters* 65, 415–417 (1994).
9. Tian, X. & Bhushan, B. The micro-meniscus effect of a thin liquid film on the static friction of rough surface contact. *Journal of Physics D: Applied Physics* 29, 163–178 (1996).
10. Hasz, K., Ye, Z., Martini, A. & Carpick, R. W. Experiments and simulations of the humidity dependence of friction between nanoasperities and graphite: The role of interfacial contact quality. *Physical Review Materials* 2, 126001 (2018).
11. Archard, J. F. The temperature of rubbing surfaces. *Wear* 2, 438–455 (1959).
12. Khare, R., de Pablo, J. & Yethiraj, A. Molecular simulation and continuum mechanics study of simple fluids in non-isothermal planar Couette flows. *The Journal of Chemical Physics* 107, 2589 (1997).

Reviewer #2 (Remarks to the Author):

This paper entitled "Graphite lubrication of highly loaded contacts: humidity-dependent friction and structural transition to turbostratic carbon", the friction and structure evolution of graphite under different load and humidity were systematically studied by microtribometer experiment and DFTB simulation. Experimental observations show that the graphite structure changes to t-C structure under load, and FeC layer is formed under high load. The simulation results show that the friction evolution process of the system is different under different loads and humidity, such as cold welding, few bonds, dry passivation and water film, which have a significant impact on the friction coefficient of the system. Their work refines graphite lubrication mechanisms and extends existing "adsorption model". However, several issues should be addressed, before the consideration of publication.

1. The microtribometer experiments (FIG. 2d) shows that under high humidity (RH 45%), the wear coefficient increases significantly, which is a interesting phenomenon. However, the following part only

gives a brief explanation of that conclusion. It is suggested to further study the causes of high wear rate in high humidity environment.

2. The marked areas in FIG. 3 represent regions of interest (ROI), and 3 circles exist in FIG. 3. However, four regions (unworn region; 5%RH 50MPa; 5%RH 1GPa and 24%RH 1GPa) are used for a preparation of lamellae that are analyzed in consecutive TEM characterization. The mark is missing in the FIG. 3b (the system of <5%RH 1GPa), and it is suggested to indicated the ROI in FIG. 3b.

3. In order to facilitate the understanding of the structural evolution of graphite samples, it is necessary to clearly mark the corresponding position relationship between BF-TEM(upper half) and HR-TEM(lower half) in FIG. 4.

4. Try to unify the scales in the same picture. For example, the scale in FIG. 4f is 60 nm (different from FIG. 4b and FIG. 4j), and the scale in FIG. 6a is 50 nm (different from FIG. 6.b and FIG. 6.b).

5. In FIG. 5d, the graphite structure on the iron surface and t-C structure far away from the iron position can be observed clearly. However, the FIG. 5c is fuzzy and all the structures seem to be amorphous, which does not reflect the gradient nature of sp² evolution in FIG. 6b.

6. The sample with humidity of 24%RH was taken in the microtribometer experiment (FIG. 4i-l), but the structure evolution of this sample was not explained in detail in the subsequent analysis. In the experiment part, the influence of humidity on the friction and structure evolution of graphite was not studied in depth.

7. In FIG. 9, the upper and lower friction pairs in the simulation are constructed as commensurate structure. Why is the friction pair constructed as a commensurate structure, and is it consistent with the actual situation.

8. In Figure 10, it is suggested to supplement the identification criteria of the four systems (cold welding; few bonds; dry passivation and water film) in the simulation process.

9. There are some description errors in the article, for example:

1) Page 3, 9 paragraph. "when reports emerged that friction and wear of graphite contacts increase drastically at low humidity." It should be rewritten as "friction and wear of graphite contacts decrease drastically at low humidity".

2) Page 5, 92 paragraph. "Notably, an experiment with a reduced Hertz pressure of 50 MPa at RH≤5 % results in a CoF of 0.28 (green circle in Fig. 2b)." It should be rewritten as "(green circle in Fig. 2c)".

3) Page 8, 124 paragraph. "One of it from an unworn region as a reference, two after experiments at 1 GPa and an additional lamella from from the low load sample testet at ≤5 % RH and 50 MPa (marked as white ovals in Fig. 3c)." It should be rewritten as "(marked as white ovals in Fig. 3a)".

10. There are illustrations in the article that are not clearly described, for example:

1) FIG. 2b. Working condition not shown (experimental load not shown).

2) FIG. 2b. Legend does not indicate the meaning of the red circle.

3) FIG. 5. The working condition is not specified (the humidity of the experimental environment is not specified).

4) FIG. 9f. Does not indicate the initial conditions of the simulation.

Rebuttal letter

Dear Reviewers,

Thank you for your valuable comments and suggestion which were quite helpful to improve our manuscript. A point-by-point response for each comment is prepared and presented below. All the changes are highlighted in blue in the revised manuscript. In this major revision, we report additional experiments that further elucidate the origin of high friction and wear at elevated relative humidity. Additional DFTB simulations have been performed to elucidate the effect of misorientation of terminal loops on top of two sliding graphite crystals.

Reviewer #1:

Reviewer: It has been known for several decades that graphite friction and wear decrease significantly at high relative humidity. At low loads, this has been attributed (amongst other potential mechanisms) to the formation of water layers adsorbed on top of the graphite, which act as boundary lubrication films (adsorption model). Here, the authors investigate using microtribometry and quantum molecular dynamics simulations whether this mechanism also prevails at high loads, which are relevant to many machine components.

The authors confirm that the high to low friction transition with increasing relative humidity also occurs for graphite at high loads. Transmission electron microscopy (TEM) shows a transition of the polycrystalline graphite lubricant into turbostratic carbon (t-C) after high (1 GPa) and even after low load (50 MPa) sliding. Quantum (DFTB) molecular dynamics simulations relate high friction and wear to cold welding and shear-induced formation of t-C. Low friction is maintained when molecular water films are preserved, even on t-C surfaces.

The manuscript is well-written, the methods are robust, and the results should be of interest to the tribology community. However, I am not sure whether it is of sufficient general interest for this journal given that the results are not particularly surprising. The results effectively confirm that the previously proposed adsorption model is applicable over a wide load range, but that stress-induced transformation of graphite to t-C at the sliding interface should also be accounted for. Some specific comments and recommendations are given below.

Answer: We thank the reviewer for acknowledging the quality of our manuscript and the robustness of our methods. We agree with her/him that the results are of interest for tribologists. However, we believe that not only specialists will benefit from our work, but also many users of graphite lubricants (e.g. to lubricate their locks) might be interested in graphite's lubrication mechanisms. Since graphite can be considered the prototypical solid lubricant, we trust that many practitioners would like to know how it works.

Reviewer: 1. Introduction: other proposed mechanisms of graphite lubrication, e.g. roller mechanism,[1,2] could also be discussed here. Indeed, this mechanism could be related to the transformation of graphite to t-C.

Answer: We thank the reviewer for pointing out this mechanism and the corresponding references. We include it in the introduction of the revised manuscript for the sake of completeness. No evidence for the relevance of this mechanism could be found in the post-mortem analysis of the wear tracks after our experiments (e.g. no graphene nanoscrolls on top or inside the t-C, as well as outside the wear track).

Text added:

"Bollmann and Spreadborough proposed a "roller mechanism" [14]: When moved laterally, individual packets of graphene layers roll together to form graphene scrolls that lubricate via a bearing-like mechanism (Fig. 1c). Since the influence of humidity is explained by water intercalation (similar to Rowe [11]), the roller model is in disagreement with later experimental X-ray characterization as well [12]."

Reviewer: 2. Introduction: The schematics could be made more visually appealing, particularly Figure 1. For example, a 3D rendering showing water molecules, fragments, surfaces etc.

Answer: We went carefully over all figures and checked their appeal. In the case of Figure 1, all authors like the simplicity and clarity of the 2D schematics. We are not sure if a 3D rendering would help the readers in a better understanding of the various models. However, the reviewer's need for more visually appealing illustrations inspired us to revise Figure 1. It now contains more detailed 2D drawings as well as a sketch of the roller model (suggested in point 1). Revised figure 1:

Reviewer: 3. Results: It should be noted here that t-C is commonly formed from the mechanical milling of graphite,[3] which also applies sliding forces (shear stresses). It is therefore perhaps unsurprising that this also occurs inside a tribological contact.

Answer: Ball milling is a technique that has been specifically designed to initiate mechanochemical reactions by severe impacts of balls within a powder. The reviewer is correct that t-C is formed under such extreme circumstances. However, for us it really came as a surprise that t-C was formed under the much milder conditions of tribological contacts in bearings.

We added a brief discussion at the end of the article:

“The transformation of graphite to t-C has been reported previously [33–35] but only during milling of graphite, where more severe conditions prevail. Thus, our research shows for the first time that sliding triggers t-C formation.”

Reviewer: 4. Results: While the TEM images provide some evidence of transformation to t-C, further surface analysis, e.g. Raman spectroscopy or XRD could be used to confirm this. This is important since it is one of the central claims of the paper.

Answer: Initially, Raman analyses of wear tracks were performed in order to quantify the evolution of sp^2 hybridization. However, later we realized that EELS measurements provided much more accurate and more local sp^2 hybridization data (see Figure 6).

Of course, XRD is a bulk analysis method and therefore it is less useful for thin film characterization. To assess crystallinity of the graphite lubricant, additional FFT analyses of the HRTEM data has been conducted for the revised manuscript. The FFT method is more suitable than XRD and supports our conclusion about the crystalline to t-C transition. We show an example in the SI.

Material added to the SI: Figure S6

Figure S6: Fast Fourier Transformation (FFT) of the HR-TEM picture taken after the experiment at 1 GPa and $\leq 5\%$ RH, (a) HR-TEM image with marked regions of interest (ROIs), (b) ROI1 close to substrate, (c) ROI2 in the middle, and (d) ROI3 close to sliding interface. The halos become more diffused when going from the substrate to the sliding interface, representing the transformation from polycrystalline graphite to t-C and its gradient when going from the substrate towards the sliding interface.

Reviewer: 5. Results: How are the selected water coverages justified compared to experiments? See also comment 8.

Answer: As correctly pointed out by the reviewer in point 9, the pressure-viscosity coefficient of water is essentially vanishing and consequently from a continuum perspective fluid entrainment will not lead to micrometer thick lubricating films under EHL conditions. Currently, work is in progress to study the entrainment of water into nanoscale a-C contacts using a non-reactive all-atom force field. Our preliminary results agree with the pioneering work of Martini and Carpick (ref 10 in the reviewers list) providing evidence that at maximum a couple of water monolayers are inside the EHL contacts. Therefore, DFTB calculations with water densities corresponding to less than 3 monolayers are sufficient to explore the lubrication regimes. For the maximum amount of water (32 molecules \approx 2.5 monolayers) all simulations were in the water layer regime (even at pressures of the order of 5 GPa) and simulations with an increased number of water molecules wouldn't provide any additional information.

A brief discussion of the comparability between the water coverages in our DFTB simulations and the experimental situation is provided in the discussion section of the revised manuscript (see also our answer to comment 9).

“We would like to point out the difficulty in establishing a direct relationship between the relative humidity in our microscale tribometer experiments and the thickness of water films in our nanoscale sliding simulations. Experiments suggest that closed water films are present on HOPG surfaces at relative humidity as low as 10% [33]. In Grand Canonical Monte Carlo simulations of an unloaded nanoscale ta-C tip sliding on HOPG several monolayers of water were observed in the nanocontact for RH>40% [34]. In principle, corresponding simulations could be performed for our graphite lubricant to relate RH with the water film thickness in our experimental microscale contacts. However, the size of the contact area renders such simulations extremely challenging. This would exceed the scope of this article and thus, we are limited to a qualitative discussion. In any case, it can be safely concluded that high pressures in combination with surface roughness and low sliding speeds result in submonolayer water films that promote cold welding and t-C formation.”

Reviewer: 6. Discussion: Other studies that have compared the tribology of graphite to t-C coatings could also be mentioned here.[4-6]

Answer: Thank you for the hint. We added reference #4 to the revised manuscript but omitted #5 and #6 as they focused more on the pretreatment of the sample with temperature and the influence of the normal force, which is not that relevant for this paper.

“Recently, the tribological performance of multilayer graphene [36] or t-C [37] in high pressure experiments and ambient conditions has been investigated. Kumar et al. observed similar trends in terms of decreasing friction with increasing humidity, but measured higher overall friction. Bhowmick et al. investigated multi-layer graphene [37]. These authors found similar trends regarding the formation of amorphous carbon, although the amorphization took only partially place producing graphene stacks embedded into the amorphous material without a height gradient in the sp^2 hybridization. In our work, a sp^2 gradient was evident both from HR-TEM images as well as quantitative EELS analyses.”

Reviewer: 7. Discussion: Very similar observations to those made in the current study (although with slightly different interpretations) have recently been made using multilayer graphene rather than graphite as a lubricant (on a different metal substrate).[7] This previous study should be discussed here.

Answer: We thank the referee for bringing the article by Bhowmick et al. to our attention. We have added a corresponding text passage to the discussion (see response to previous point).

Reviewer: 8. Discussion: A more detailed explanation of the increase in friction and wear at very high relative humidity (capillary condensation) should be provided. Generally speaking, capillary condensation only has a significant effect on friction for hydrophilic surfaces.[8] Since graphite is hydrophobic, this supports the authors claim that the interface is passivated by polar groups that increase its hydrophilicity. Most measurements of the hydrophobic effect are from nanoscale single asperity studies using atomic microscopy. However, the authors should attempt to quantify the potential contribution of capillary effects to adhesion and friction in this system. Some methods to estimate the contribution of capillary condensation to friction in multi-asperity contacts are given in Ref. [9].

Answer: We are grateful to the reviewer for bringing up this issue. The reviewer's remark motivated us to conduct an additional microtribometer experiment, where the contact is flooded by a droplet of water. Although the formation of capillary necks can be excluded for this experimental protocol, we still observe high wear.

This observation agrees with an estimate of the effect of capillary necks on the normal force under moist conditions. The additional normal force due to capillary condensation can be estimated using $F_c = 4 \pi \gamma R \cos \theta$. Assuming a mildly hydrophilic graphite surface ($\theta \sim 60^\circ$), a surface tension of water $\gamma = 72.8 \cdot 10^{-3} \frac{N}{m}$ and a tip radius $R = 1 \text{ mm}$ the capillary force F_c accounts for 0.1% of the externally applied normal force. Thus, the intuition of the reviewer was correct. The experiment with the fully flooded contact as well as the estimate of the capillary contribution to the normal force indicate that there must be another reason for the observed high wear.

As capillary condensation can be ruled out, the only alternative explanation for the high wear that comes to our mind consists of an instability of the coating with respect to water. Water is a strong solvent. Most likely the water molecules diffuse into and underneath the coating leading to decomposition or delamination of the graphite solid lubricant.

To corroborate this mechanism, we ultrasonicated a graphite-coated sample in a water bath. Indeed, already after a few seconds large patches of the coating dissolved from the substrate and were subsequently removed (see Fig. S4). The revised manuscript reports the new experimental data accompanied by a detailed discussion. **The remarks and calculations regarding the capillary forces have been shifted to the SI.**

“As a working hypothesis, an excess of water on the graphite lubricant could promote dissolution of the graphite crystals or even a delamination of the coating. Above a certain threshold (here $\geq 45\%$ RH) water might diffuse into and especially underneath the graphite coating in larger quantities, hence forming internal water films. As a consequence, the lubricant would be removed during the first sliding cycles, as observed in our experiment. To corroborate this scenario, additional microtribometer experiments are conducted where the sphere was immersed into a drop of water placed on the

graphite-coated plate; hence the effect of capillary condensation can be ruled out. As evident from Fig. 2b the CoF immediately rises to high values after a few cycles and exhibited high wear (see Fig. 2d) which hints to a quick removal of the coating.

As a further test of our dissolution scenario, a graphite-coated plate was completely immersed into deionized water and put into an ultrasonication bath. Already after a few seconds delamination of the coating is observed resulting in its complete removal after 1 min, as evident from Figure S4. This suggests that the observed increased wear at high relative humidity is a consequence of the dissolution of the coating by water.”

Reviewer: 9. Methods: In the DFTB calculations, the number of water molecules is varied between 1 and 32, which is supposed to represent different relative humidity values in the experiments. In previous classical MD simulations of graphite friction in the presence of water, Grand Canonical Monte Carlo calculations have been used to explicitly link relative humidity to graphite surface coverage.[10] While I recognise that GCMC calculations would be prohibitively expensive using DFTB calculations, performing such calculations for the current systems using classical MD would enable a more robust link between the relative humidity in the experiments and the number of water molecules in the simulations. Note that given the very low pressure-viscosity coefficient of water, fluid entrainment (and its load dependence) will be negligible for the current systems. Thus, elastohydrodynamic lubrication analysis (as suggested as a potential solution in the Discussion section) is unlikely to give an accurate estimate of the water coverage at the sliding interface.

Answer: We agree with the reviewer that it would be desirable to establish a direct link between the number of water molecules on the surface and the experimental humidity. Simulations using a Grand Canonical Monte Carlo method, as suggested by the reviewer, would indeed be one way to achieve this, however the reviewer has already correctly pointed out that this is not possible with DFTB simulations due to the high computational cost. Simulations using classical non-reactive molecular dynamics would be an alternative, which would allow significantly larger systems and longer simulation times but it still would be a challenge to get close to the experimental time and length scales. We adopt the reviewer's comment as a suggestion for future simulations, but we believe that such investigations are beyond the scope of this paper. A brief discussion of this issue and a reference to the reviewer's reference [10] have been added to the manuscript (see also our answer to comment 5).

Furthermore, we thank the reviewer for his/her comment on elastohydrodynamic simulations. The reviewer is correct in his judgment that our originally proposed simulations for water would be of limited informative value. The relevant part of the discussion has been removed.

Reviewer: 10. Methods: Are the tribology experiments expected to be isothermal? If not, please provide an estimate of the amount of frictional heating (using Archard equation [11] or similar).

Answer: Using Blok theory we estimated a maximal temperature increase of 5 mK for the 1 GPa experiments and 0,28 nK for the 50 MPa experiments for boundary lubrication of the steel-iron contact. The details are added to the SI.

We added to the methods section:

“An estimation of the flash temperature as well as the capillary force was conducted and can be found in the SI file. Increase of temperature due to sliding as well as normal forces due to capillarity is negligible.”

Reviewer: 11. Methods: As is common when comparing molecular simulations to experiments, the sliding velocity in the DFTB simulations (100 m/s) is much faster than in the associated experiments (0.5 mm/s). The friction behaviour cannot reasonably be expected to be independent of sliding velocity over this range for the studied systems and this should be acknowledged.

Answer: We thank the reviewer for this remark. Indeed, the friction response in experiment and simulation should not be compared on a quantitative basis. However, we believe that our simulations provide useful insights into the tribo-induced transformations and corresponding friction regimes in the experiments. We added a paragraph to the methods section of the revised manuscript in which we discuss this issue.

“We would like to point out that our model systems do not allow a direct comparison with the experimental friction behavior. The high computational cost limits DFTB-MD simulations to system sizes of about 1 nm in each spatial direction and simulation times of less than 1 ns. Typical sliding velocities in the simulation are 5 orders of magnitude higher than in the experiments (100 m/s vs. 0.5 mm/s). Therefore, the frequency of thermal barrier crossings is underestimated and most reactions occur via stress-induced instabilities. However, it is evident from previous studies that simulations at higher shear rates still provide useful qualitative insights into mechanochemical reactions in carbon based tribosystems [41,42].”

Reviewer: 12. Methods: Related to the point above, thermostating the entire system to 300 K prevents thermal gradients from forming, as they do in real experiments.[12] Moreover, thermostating confined fluid molecules (water here) can artificially affect their behaviour, particularly when using a stochastic thermostat such as Langevin. Thus, it is usually preferable to thermostat only the outer layers of confined systems.

Answer: The reviewer is correct with her/his general remark that coupling fluid molecules to a Langevin thermostat affects their behavior. As mentioned by the reviewer, the usual procedure to circumvent this is to couple only some layers with a certain distance to the sliding interface to the thermostats. However, due to the high computational cost of the DFTB method employed in this study, our systems are small and are sheared at high velocities. Coupling only thin layers of atoms to the thermostats could therefore lead to an artificial increase of the system temperature. We therefore chose to thermalize the entire system. In order to minimize artefacts caused by this thermalization procedure, we did not use a conventional three-dimensional Langevin thermostat, but one that acts only perpendicular to the sliding direction and to the graphite surfaces, i.e. along the $\langle 1\bar{2}10 \rangle$ -direction of the graphite crystals. The overall interaction between the water molecules and the surfaces should therefore not be affected by the thermalization. We added a sentence to the description of our simulations to clarify this.

“A Langevin thermostat [44] at 300 K acting along the $\langle 1\bar{2}10 \rangle$ -direction of the graphite lattice only, i.e. perpendicular to the sliding direction and perpendicular to the graphite surfaces, controls the temperature of the system. This ensures efficient temperature control and simultaneously minimizes the influence of the thermostat on the interactions between the water molecules and the graphite surfaces.”

The reviewer’s references:

1. Bollmann, W. & Spreadborough, J. Action of Graphite as a Lubricant. *Nature* 186, 29–30 (1960).
2. Spreadborough, J. The frictional behaviour of graphite. *Wear* 5, 18–30 (1962).
3. Li, Z. Q., Lu, C. J., Xia, Z. P., Zhou, Y. & Luo, Z. X-ray diffraction patterns of graphite and turbostratic carbon. *Carbon*. 45, 1686–1695 (2007).
4. Kumar, N., Dash, S., Tyagi, A. K. & Raj, B. Super low to high friction of turbostratic graphite under various atmospheric test conditions. *Tribology International* 44, 1969–1978 (2011).
5. Kumar, N. et al. High-temperature phase transformation and low friction behaviour in highly disordered turbostratic graphite. *Journal of Physics D: Applied Physics* 46, (2013).
6. Kumar, N., Kozakov, A. T., Ravindran, T. R., Dash, S. & Tyagi, A. K. Load dependent friction coefficient of crystalline graphite and anomalous behavior of wear dimension. *Tribology International* 88, 280–289 (2015).
7. Bhowmick, S., Banerji, A. & Alpas, A. T. Role of humidity in reducing sliding friction of multilayered graphene. *Carbon* 87, 374–384 (2015).
8. Binggeli, M. & Mate, C. M. Influence of capillary condensation of water on nanotribology studied by force microscopy. *Applied Physics Letters* 65, 415–417 (1994).
9. Tian, X. & Bhushan, B. The micro-meniscus effect of a thin liquid film on the static friction of rough surface contact. *Journal of Physics D: Applied Physics* 29, 163–178 (1996).
10. Hasz, K., Ye, Z., Martini, A. & Carpick, R. W. Experiments and simulations of the humidity dependence of friction between nanoasperities and graphite: The role of interfacial contact quality. *Physical Review Materials* 2, 126001 (2018).
11. Archard, J. F. The temperature of rubbing surfaces. *Wear* 2, 438–455 (1959).
12. Khare, R., de Pablo, J. & Yethiraj, A. Molecular simulation and continuum mechanics study of simple fluids in non-isothermal planar Couette flows. *The Journal of Chemical Physics* 107, 2589 (1997).

Reviewer #2:

Reviewer: This paper entitled "Graphite lubrication of highly loaded contacts: humidity-dependent friction and structural transition to turbostratic carbon", the friction and structure evolution of graphite under different load and humidity were systematically studied by microtribometer experiment and DFTB simulation. Experimental observations show that the graphite structure changes to t-C structure under load, and FeC layer is formed under high load. The simulation results show that the friction evolution process of the system is different under different loads and humidity, such as cold welding, few bonds, dry passivation and water film, which have a significant impact on the friction coefficient of the system. Their work refines graphite lubrication mechanisms and extends existing "adsorption model". However, several issues should be addressed, before the consideration of publication.

Answer: We thank the reviewer for carefully reading our paper and recommending publication after addressing minor issues.

Reviewer: 1. The microtribometer experiments (FIG. 2d) shows that under high humidity (RH 45%), the wear coefficient increases significantly, which is an interesting phenomenon. However, the following part only gives a brief explanation of that conclusion. It is suggested to further study the causes of high wear rate in high humidity environment.

Answer: The reviewer asks in principle the same import question as reviewer 1. We thank the reviewer for this opportunity to improve our manuscript. To investigate this matter in more detail, we conducted an additional microtribometer experiment (see Fig.2), where a droplet of water was added into the contact. Thus, no capillary necks will be formed and indeed we still observed high wear. We therefore came up with an alternative explanation for the high wear at elevated humidity: the water seems to diffuse into and underneath the coating. In consequence, the coating will be "flooded" away upon any lateral force. To prove this mechanism further, we inserted a graphite-coated sample into water and placed it into an ultrasonication bath. As expected, already after a few seconds large patches of the coating dissolved from the substrate and were subsequently removed (see Fig. S4). We added the experimental data to the manuscript as well as a detailed discussion (see response to reviewer #1).

Reviewer: 2. The marked areas in FIG. 3 represent regions of interest (ROI), and 3 circles exist in FIG. 3. However, four regions (unworn region; 5%RH 50MPa; 5%RH 1GPa and 24%RH 1GPa) are used for a preparation of lamellae that are analyzed in consecutive TEM characterization. The mark is missing in the FIG. 3b (the system of <5%RH 1GPa), and it is suggested to indicated the ROI in FIG. 3b.

Answer: Indeed, one circle was missing. We rectified this in the revised manuscript.

Reviewer: 3. In order to facilitate the understanding of the structural evolution of graphite samples, it is necessary to clearly mark the corresponding position relationship between BF-TEM(upper half) and HR-TEM (lower half) in FIG. 4.

Answer: In the revised manuscript the approximate zoom regions of HR-TEM are marked in the BF-TEM images with blue marks to enable a better understanding. We also added a sentence to the image description:

“Marked in light and dark blue are the approximate positions of the HR-TEM studies in the BF-TEM images.”

Reviewer: 4. Try to unify the scales in the same picture. For example, the scale in FIG. 4f is 60 nm (different from FIG. 4b and FIG. 4j), and the scale in FIG. 6a is 50 nm (different from FIG. 6.b and FIG. 6.b).

Answer: Fig. 3, 4, 5, and 6 have been changed accordingly to unify the scales in the revised manuscript.

Reviewer: 5. In FIG. 5d, the graphite structure on the iron surface and t-C structure far away from the iron position can be observed clearly. However, the FIG. 5c is fuzzy and all the structures seem to be amorphous, which does not reflect the gradient nature of sp² evolution in FIG. 6b.

Answer: In Fig 5c graphene bundles are clearly perceptible. Maybe the figure sent to the reviewer had lost resolution. Now, we provide a zoom in the SI to convince the reader of our claims about the structure.

Reviewer: 6. The sample with humidity of 24%RH was taken in the microtribometer experiment (FIG. 4i-l), but the structure evolution of this sample was not explained in detail in the subsequent analysis. In the experiment part, the influence of humidity on the friction and structure evolution of graphite was not studied in depth.

Answer: For 1 GPa, we analyzed in detail all relevant humidity regimes (i.e. low and medium). For very high RH, the final wear scar was not informative. We show the medium case in the main text and moved the low humidity case to the SI. Obviously, it is easy to overlook this and therefore we make it clearer in main text:

“With the selected three experimental samples the whole range of humidity and normal force is covered. Samples at humidity $\geq 36\%$ RH were not investigated as the sample was exposed to extensive abrasive wear, thus the carbon layer structure could not be analyzed.”

Reviewer: 7. In FIG. 9, the upper and lower friction pairs in the simulation are constructed as commensurate structure. Why is the friction pair constructed as a commensurate structure, and is it consistent with the actual situation.

Answer: Due to DFTB's high computational costs the model systems are small and need to be chosen carefully. We decided to focus on loops in the upper and lower sliding body having the same directions. It is reasonable to assume that the graphite crystals in the experiments are randomly oriented in our experimental coatings. Therefore, crystals with parallel aligned loops represent certainly a limiting case. In order to explore the influence of loop orientation, we studied also perpendicular loops. These additional calculations using a crossed loop configuration show similar outcomes. Cold welding of the surfaces and the formation of water films is observed at similar water contents. However, higher normal pressures were necessary to induce cold welding of the surfaces and repassivation during sliding was not observed on the time scale of our simulations. This indicates

a reduced tendency for t-C formation if the graphite crystals are oriented in different directions. We added a remark to the main text and a comparison between the parallel and the crossed loops to the supporting information.

Reviewer: 8. In Figure 10, it is suggested to supplement the identification criteria of the four systems (cold welding; few bonds; dry passivation and water film) in the simulation process.

Answer: We thank the reviewer for pointing out this ambiguity. All trajectories were subject to a visual inspection. In combination with the characteristic profiles of the shear stresses vs. time (Fig. 9), this allowed a clear identification of the different regimes. We added a description of the identification process to the results section.

“From each trajectory a normalized probability density (PD) of the instantaneous shear stresses is obtained by histogramming the $\tau(t)$ curves (Fig. 10a). All trajectories are visually inspected. In combination with the characteristic profiles of shear stresses vs. time (Fig. 9), the different states can be identified unambiguously. Since typical shear stresses in the cold-welded, the partially passivated, the full passivated, and the water lubricated states differ significantly, these states occur in the PDs as clearly separated peaks. A set of Gaussian functions is fitted to the PD and associated with the different states based on the described visual inspection.”

Reviewer: 9. There are some description errors in the article, for example:
1) Page 3, 9 paragraph. “when reports emerged that friction and wear of graphite contacts increase drastically at low humidity.” It should be rewritten as “friction and wear of graphite contacts decrease drastically at low humidity”.

Answer: Here, we disagree with the reviewer. We checked Baker (1935) and as a matter of fact he reported an **increase** of friction and wear for decreasing humidity.

2) Page 5, 92 paragraph. “Notably, an experiment with a reduced Hertz pressure of 50 MPa at RH≤5 % results in a CoF of 0.28 (green circle in Fig. 2b).” It should be rewritten as “(green circle in Fig. 2c)”.

Answer: Thank you for the hint to this error. We corrected the reference to the figure panel.

3) Page 8, 124 paragraph. “One of it from an unworn region as a reference, two after experiments at 1 GPa and an additional lamella from from the low load sample testet at ≤5 % RH and 50 MPa (marked as white ovals in Fig. 3c).” It should be rewritten as “(marked as white ovals in Fig. 3a)”.

Answer: Thanks for pointing this out. We updated the reference to the figure panel.

Reviewer: 10. There are illustrations in the article that are not clearly described, for example:
1) FIG. 2b. Working condition not shown (experimental load not shown).

Answer: We added the load conditions to Fig. 2b.

2) FIG. 2b. Legend does not indicate the meaning of the red circle.

Answer: We added the meaning of the red circle to the figure caption of Fig. 2c.

3) FIG. 5. The working condition is not specified (the humidity of the experimental environment is not specified).

Answer: We added the humidity to Fig. 5.

4) FIG. 9f. Does not indicate the initial conditions of the simulation.

Answer: We re-organized the figure and added snapshots of the trajectory. Videos of all trajectories from which we show snapshots in the manuscript are provided as supplemental material.

REVIEWERS' COMMENTS

Reviewer #1 (Remarks to the Author):

I thank the authors for addressing most of my comments, I now recommend publication in the current form.

Reviewer #2 (Remarks to the Author):

The authors have properly addressed my previous concern, I do not have other questions now.